

# Finding self-similar behavior in quantum many-body dynamics via persistent homology

Daniel Spitz[1*], Jürgen Berges[1], Markus Oberthaler[2] and Anna Wienhard[3, 4]

**1** Institut für theoretische Physik, Ruprecht-Karls-Universität Heidelberg,
Philosophenweg 16, 69120 Heidelberg, Germany
**2** Kirchhoff-Institut für Physik, Ruprecht-Karls-Universität Heidelberg,
Im Neuenheimer Feld 227, 69120 Heidelberg, Germany
**3** Mathematisches Institut, Ruprecht-Karls-Universität Heidelberg,
Im Neuenheimer Feld 205, 69120 Heidelberg, Germany
**4** HITS gGmbH, Heidelberg Institute for Theoretical Studies,
Schloss-Wolfsbrunnenweg 35, 69118 Heidelberg, Germany

⋆ spitz@thphys.uni-heidelberg.de

## Abstract

Inspired by topological data analysis techniques, we introduce persistent homology observables and apply them in a geometric analysis of the dynamics of quantum field theories. As a prototype application, we consider data from a classical-statistical simulation of a two-dimensional Bose gas far from equilibrium. We discover a continuous spectrum of dynamical scaling exponents, which provides a refined classification of nonequilibrium self-similar phenomena. A possible explanation of the underlying processes is provided in terms of mixing strong wave turbulence and anomalous vortex kinetics components in point clouds. We find that the persistent homology scaling exponents are inherently linked to the geometry of the system, as the derivation of a packing relation reveals. The approach opens new ways of analyzing quantum many-body dynamics in terms of robust topological structures beyond standard field theoretic techniques.

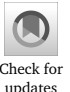

# 1  Introduction

Over the past two decades the mathematical field of topological data analysis (TDA) has gained considerable attention, accompanied by far-reaching theoretical and computational developments [1, 2]. Prominently, with the notion of persistent homology the TDA toolbox offers a versatile and numerically fairly simply applicable tool to study topological features contained in data, such as connected components, loops or voids [3–5]. In particular, persistent homology associates length scales to such topological features, allowing for a numerical discrimination of dominant features and possible noise in data. To accomplish this, simplicial complexes such as so-called Čech complexes, Vietoris-Rips complexes or alpha shapes [6, 7] are employed. Besides the mathematical investigations on persistent homology, very fruitful

applications to physical systems include studies in astrophysics and cosmology [8–11], physical chemistry [12], amorphous materials [13], quantum algorithms [14–18] and the theory of quantum phase space [19]. In particular, persistent homology has been successfully applied to the detection of equilibrium phase transitions in statistical mechanics [20] as well as to the identification of phases in lattice spin models [21].

In this work, we propose persistent homology observables for the analysis of the dynamics of quantum many-body systems. As a prototype application, we consider a Bose gas far from equilibrium. While there are many different ways of driving a Bose gas away from equilibrium, it has recently been demonstrated experimentally that the subsequent relaxation dynamics can exhibit universal properties that are insensitive to the details of the initial conditions and system parameters [22–24]. Theoretical results based on field correlation functions indicate that vastly different systems far from equilibrium may share very similar universal scaling properties, ranging from post-inflationary dynamics in the early universe [25, 26], and ultra-relativistic collision experiments with heavy nuclei [27–29], to ultra-cold quantum gases in the laboratory [30, 31]. In particular, quantum as well as classical statistical field theories appear to belong to the same nonthermal universality class [32]. These similarities have to be tested against refined analysis and classification schemes. We will exploit the multi-scale topological information encoded in a family of alpha complexes and in associated persistent homology groups in order to analyze self-similar scaling dynamics in position space variables.

More precisely, serving as a numerical testbed, we apply TDA techniques to the dynamics of the single-component nonrelativistic Bose gas in two spatial dimensions, described by the time-dependent Gross-Pitaevskii equation with quantum initial conditions. The latter exhibits a rich phenomenology far from equilibrium, including various nonthermal fixed points associated to regimes of weak and strong wave turbulence [33–35]. Focussing on the nonperturbative strong wave turbulence regime, a vertex-resummed two particle-irreducible expansion scheme has been successfully employed to obtain analytical predictions for relevant scaling exponents [32, 36]. The existence of corresponding nonthermal fixed points has been confirmed by means of numerical lattice simulations [37]. In addition, the infrared nonthermal fixed point can be dominated by vorticial excitations interacting anomalously with each other via 3-vortex interactions [37, 38], that is, altering the universal scaling behavior. It has been conjectured that this anomalous vortex kinetics is associated to the formation of Onsager vortex clusters out of equilibrium via evaporative heating [39, 40]. Recently, experimental evidence for scale-invariant dynamics and Onsager's model has been reported [41, 42].

Guided by numerical results for the two-dimensional Bose gas, we reveal that at late times far from equilibrium persistent homology observables can show self-similar scaling characteristic to a nonthermal fixed point. We discover a continuous spectrum of dynamical scaling exponents, depending on a filtration parameter to construct point clouds, which provides a refined classification of nonequilibrium self-similar phenomena. The existence of such a scaling exponent spectrum seems to indicate scaling species mixing, in our case between the strong wave turbulence and the anomalous vortex kinetics nonthermal fixed points present in the infrared of the particular Bose gas. The analysis is supplemented by a thorough investigation of accompanying subtleties of the chosen persistent homology approach such as amplitude redistribution-induced exponent shifts.

On the theoretical side, we define persistent homology observables. We introduce the notion of a persistence pair distribution and its statistical asymptotics in order to infer self-similar behavior of the latter. We reveal that the appearing scaling exponents probe the geometry at hand, as indicated by a packing relation heuristically derived in this study.

This publication is structured as follows. We first describe the lattice simulations and discuss self-similar scaling for the occupation number spectrum in Sec. 2. With the Bose gas simulations at hand, we introduce and study point clouds and persistent homology groups in

Sec. 3. Rediscovering self-similarity, this exploration culminates in the existence of a scaling exponent spectrum. In Sec. 4 we carry out the construction of persistent homology observables in the classical-statistical framework, introduce the asymptotic persistence pair distribution and related geometric quantities and investigate a corresponding self-similar scaling ansatz. We discuss amplitude redistribution-induced exponent shifts, persistences and Betti number distributions in Sec. 5. Finally, in Sec. 6 we summarize, draw conclusions and issue an outlook.

# 2 Self-similarity in occupation numbers

Laying the foundations for the introduction of persistent homology observables, we first discuss self-similar scaling in the two-dimensional Bose gas for the well-established occupation number spectrum. The two-dimensional Bose gas is among the simplest systems to give rise to different nonthermal fixed points and to allow for the fast and reasonable[1] computation of persistent homology observables. We start this section by introducing the lattice simulations.

## 2.1 Simulation prerequisites

The nonrelativistic Bose gas can be described by complex scalar fields $\psi(t, \mathbf{x})$ depending on time and space, in numerical simulations restricted to a spatial lattice and time-evolved in discrete time-steps. We focus on the overoccupied regime, in which the classical-statistical approximation is suitable [32]. Accordingly, at initial time $t = 0$ a number $k$ of classical field configurations is sampled from a Gaussian ensemble, computing their individual subsequent dynamics according to the time-dependent Gross-Pitaevskii equation as described in Appendix E. In the classical-statistical approximation expectation values of an observable are computed as ensemble-averages of the observable evaluated for individual field configurations.

Given a field configuration $\psi(t, \mathbf{x})$, we define the statistical two-point correlation function

$$F(t, t', \mathbf{x} - \mathbf{x}') = \frac{1}{2} \langle \psi(t, \mathbf{x}) \psi^*(t', \mathbf{x}') + \psi(t', \mathbf{x}') \psi^*(t, \mathbf{x}) \rangle, \tag{1}$$

$\langle \cdot \rangle$ indicating evaluating the expectation value in the classical-statistical ensemble. Subsequently, with momentum denoted by $\mathbf{p}$ we define the occupation number spectrum $f(t, \mathbf{p})$ via

$$f(t, \mathbf{p}) + (2\pi)^3 \delta^{(3)}(\mathbf{p}) |\psi_0|^2(t) \equiv \int d^3x \, e^{-i\mathbf{p}\mathbf{x}} F(t, t, \mathbf{x}). \tag{2}$$

Due to spatial isotropy of expectation values in the system, the distribution function only depends on the modulus of momenta, $f(t, p) \equiv f(t, |\mathbf{p}|)$. The term $\sim |\psi_0|^2(t)$ represents a condensate occurring in the system.

We choose the initial occupation number spectrum to describe overoccupation up to a characteristic momentum scale $Q$. To this end, initial field configurations are defined as

$$f(0, \mathbf{p}) = f_0 \Theta(Q - |\mathbf{p}|), \tag{3}$$

with $f_0 = 50/(2mgQ)$ in the simulations. Unlike a system in thermal equilibrium, where the typical occupancy is of order unity at a characteristic temperature scale $T$, here we consider a nonequilibrium system where the occupancy at a given characteristic scale $Q$ is much higher than unity. Any dimensionful physical quantity will be given in units of $Q$. We set the mass $m/Q = 8$ and coupling $Qg = 0.0625$ throughout this work. Outside the box, no 'quantum-half'

---

[1]Persistent homology groups of point clouds in one spatial dimension describe connected components present in the data. In two spatial dimensions, topologically more interesting loop-like structures can be studied.

is taken into account and no initial condensate is specified. Spatial coordinates are restricted to a square lattice, $\Lambda$, consisting of a regular grid of $N^2$ points within a volume $L^2$ with periodic boundary conditions. Throughout this work, the lattice spacing reads $Qa = 0.0625$, the number of lattice sites $N = 1536$, such that

$$\Lambda = \{(an_1, an_2) \,|\, n_1, n_2 \in \{0, \dots, N-1\}\}. \tag{4}$$

If not stated differently, we average over $k = 72$ classical-statistical realizations to compute classical-statistical expectation values. For further details on the numerical simulations we refer to Appendix E.

## 2.2 Self-similarity in the occupation number spectrum

After a relatively short time interval with a quick redistribution of the initial mode occupancies, the dynamics slows down and begins to indicate the vicinity of a nonthermal fixed point by means of self-similarity. Self-similar scaling of the occupation number spectrum $f(t, \mathbf{p})$ is described by a scaling ansatz including two scaling exponents, $\alpha$ and $\beta$,

$$f(t, p) = (t/t')^{\alpha} f(t', (t/t')^{\beta} p). \tag{5}$$

In the infrared regime, a thorough numerical analysis as described in Ref. [32] yields the following scaling exponents,

$$\beta = 0.189 \pm 0.011, \qquad \alpha = 0.395 \pm 0.025, \tag{6}$$

choosing reference time $Qt' = 1250$, fitting momenta between $p/Q = 0.07$ and $p/Q = 0.7$ and times between $Qt = 1875$ and $Qt = 37500$. Thus, $\alpha/\beta = 2.09 \pm 0.18$. In Fig. 1 occupation number spectra are displayed in the infrared regime. By means of the residuals the correctness of the extracted scaling exponents can be easily verified.

The results confirm the findings for box initial conditions in Ref. [38], in which the infrared dynamics of a two-dimensional relativistic scalar field theory has been mapped to that of nonrelativistic complex scalar fields. The extracted scaling exponent $\beta$ is in very good agreement with the prediction for the anomalous vortex kinetics nonthermal fixed point in a nonrelativistic single-component Bose gas, attributed to the specific dynamics of vortex defects and related vortex interactions [37]. Additionally, $\alpha/\beta \approx 2$ indicates the transport of particle numbers to lower momenta [32].

# 3 Persistent homology in a Bose gas

Given the lattice simulations of the nonrelativistic Bose gas described in the previous section, we introduce a simple approach to construct point clouds from field configurations, namely as sublevel sets of field amplitudes. A rather intuitive sketch of the construction of alpha complexes and persistent homology groups from such point clouds is provided. In corresponding far-from-equilibrium simulations we discover growing geometric structures and self-similar scaling at large length scales. In particular, the existence of a scaling exponent spectrum is revealed. By means of the mixing of scaling dynamics species we offer a possible route to explain this finding.

## 3.1 Phenomenology of point clouds

Given a classical-statistical field realization $\psi(t, \mathbf{x})$, an immense freedom of choice exists in constructing point clouds, which are, generally speaking, finite sets of points in an arbitrary

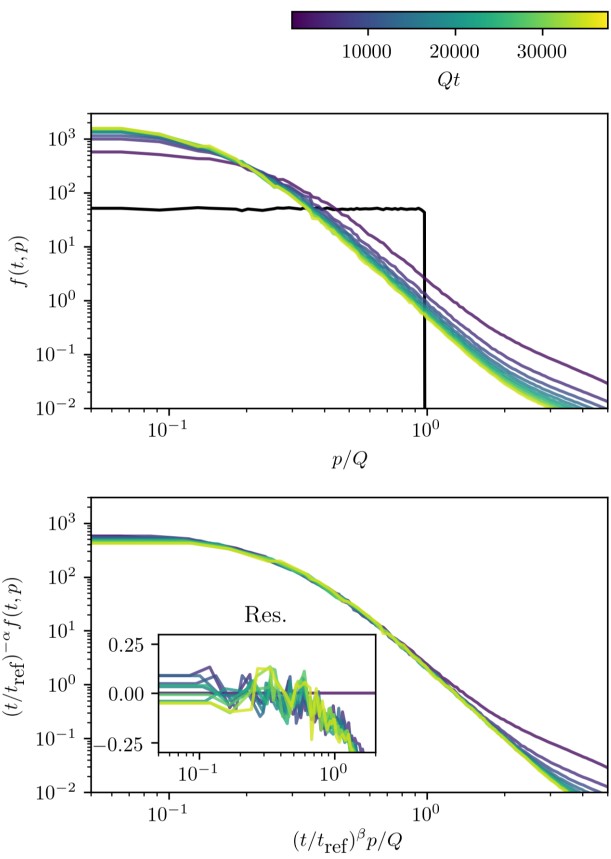

Figure 1: Occupation number distributions in the infrared. In black: The initial unrescaled occupation number distribution.

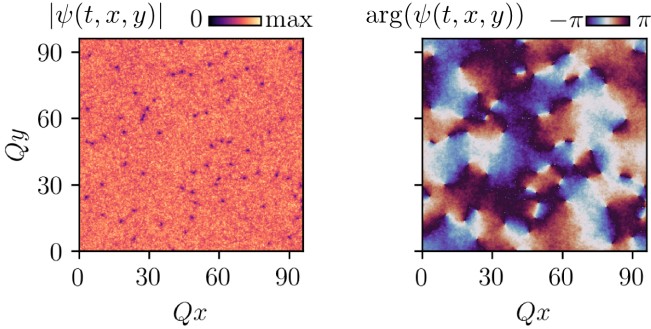

Figure 2: Amplitudes (left) and phases (right) of an example field configuration at time $Qt = 3750$.

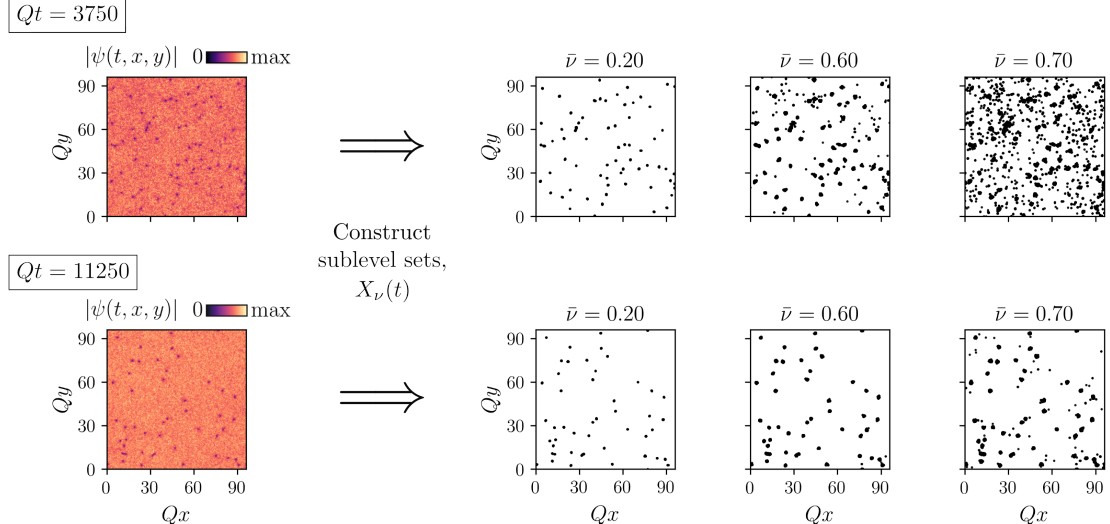

Figure 3: Amplitudes of an example field configuration and corresponding point clouds. First column from the left: Spatially-resolved field amplitudes, $|\psi(t,x,y)|$. Second to fourth column: Point clouds $X_\nu(t)$ for the different $\bar{\nu}$-values indicated. First row: $Qt = 3750$. Second row: $Qt = 11250$.

Euclidean space. We define a *filtration function* $f$ to be a map from $\mathbb{C}$ to $\mathbb{R}$ used to generate point clouds as subsets of the lattice $\Lambda$. We may construct point clouds as sublevel sets of $f(\psi(t,\cdot))$, that is, at time $Qt$ define them as $\{\mathbf{x} \in \Lambda \,|\, f(\psi(t,\mathbf{x})) \in (-\infty, \nu]\}$ for a *filtration parameter* $\nu$. In this work, point clouds are generated as sublevel sets of the field amplitude, thus defining

$$X_\nu(t) := \{\mathbf{x} \in \Lambda \,|\, |\psi(t,\mathbf{x})| \leq \nu\}. \tag{7}$$

By means of this definition, the ensemble of classical-statistical field realizations translates for each time $Qt$ into an ensemble of point clouds. Numerically, we specify the filtration parameter $\nu$ by means of the dimensionless variant $\bar{\nu}$,

$$\bar{\nu} := \nu / \langle |\psi(t=0)| \rangle_{\text{vol}}, \tag{8}$$

with the volume-averaged initial field amplitude

$$\langle |\psi(t=0)| \rangle_{\text{vol}} = \frac{1}{N^2} \sum_{\mathbf{x} \in \Lambda} |\psi(t=0, \mathbf{x})|. \tag{9}$$

   We want to emphasize that in experiments with cold atoms optical density images as given by the square of the amplitudes displayed in Fig. 2 and used in the filtration protocol, Eq. (7), form a typical observational quantity and can be easily accessed via absorption images. Varying the filtration parameter $\bar{\nu}$ amounts to measurements up to the square root of corresponding condensate densities, highlighting the physical significance of the employed point cloud construction via Eq. (7).

   Simulating on a spatial square lattice with constant lattice spacing, we want to stress that to obtain finite point clouds by means of Eq. (7) the finiteness of the lattice is crucial. Else, $X_\nu(t)$ might consist of infinitely many points. The subsequent construction of persistent homology observables, described in detail in Sec. 3.2, is robust against perturbations of the lattice points[2]. This renders the microscopic form of the lattice irrelevant for later numerical persis-

---

[2]Mathematically speaking, in a number of ways persistent homology groups are stable against perturbations of corresponding input, cf. inter alia Refs. [43, 44]. This implies, that if points in $X_\nu(t)$ are altered slightly, then persistence diagrams of the sequence of alpha complexes of $X_\nu(t)$ change only slightly, too.

tent homology results. The constant lattice spacing and finite lattice volume solely amount to a smallest and a largest length scale amenable to the investigated real-time dynamics.

In Fig. 2 amplitudes and phases of a single classical-statistical field realization are displayed. One may first note from the amplitudes on the left that in position space the system comprises two major components: fluctuations in the bulk around a mean amplitude value larger than zero and distinct minima with minimum values near to zero. While phases differ locally only slightly in regions where minima are absent, around each minimum phase windings with shifts of $\pm 2\pi$ occur. Thus, the minima can be identified with elementary vortex nuclei.

In Fig. 3 at two different times we show spatially-resolved amplitudes and a variety of point clouds computed from a single classical-statistical field realization. In point clouds $X_\nu(t)$ as defined by Eq. (7), at both times visualized we find clear manifestations of the aforementioned two components appearing in amplitudes. Having approximately zero amplitude at the center of their nuclei, vortices dominate the point clouds $X_\nu(t)$ for small filtration parameters such as $\bar\nu = 0.2$. In the limit of $\bar\nu \to 0$ point clouds actually comprise mostly vortex positions themselves, although the presence of points originating from bulk density fluctuations cannot be excluded. Described by point vortex models, for this reason the low-$\bar\nu$ limit can be associated to the incompressible limit of the theory. Increasing $\bar\nu$, in point clouds points first accumulate around vortex nuclei but at moderately high values such as $\bar\nu = 0.6$ also occur in the bulk. The higher $\bar\nu$ gets, the denser point clouds become, reducing the average distance between points. Hence, studying point clouds at different $\bar\nu$-values effectively probes the system on different length scales.

Comparing the two times displayed, we note that the number of vortices decreases with time, or, equivalently, the average inter-vortex distance increases. In Fig. 3 point clouds at $\bar\nu = 0.2$ reflect this behavior, becoming sparser in the course of time. Similarly, at higher values of $\bar\nu$ the density of points in point clouds decreases in regions where vortices are absent. All this indicates that in the temporal regime of the displayed times geometric structures in point clouds continuously grow at large length scales.

Yet, one may notice that for $\bar\nu = 0.6$ and $\bar\nu = 0.7$ the number of points in the bulk decreases faster compared to the decline in vortex numbers. This provides a first hint at the presence of different components, whose dynamics differ in terms of "speed".

## 3.2 An introduction to persistent homology

To obtain a robust quantitative means of the topological structure present in a point cloud $X_\nu(t)$, persistent homology can be employed. Aiming at an intuitive treatment, with a point cloud $X_\nu(t)$ at hand as it appears in the Bose gas simulations we introduce relevant notions from computational topology. From given input data we first define the Delaunay complex and a notion of the size of a simplex. The so-called Delaunay radius function can then be used to construct a nested sequence of subcomplexes, called alpha complexes, whose persistent homology groups form our objects of interest and eventually provide multi-scale information on the topological structure of the input point cloud. While we carry out constructions in two spatial dimensions here, they generalize easily to higher dimensions.

In Appendix A we rigorously introduce relevant fundamental algebraic topology notions and discuss the mathematical construction of persistent homology groups. For a general introduction to algebraic topology we refer to Ref. [45]; for a thorough introduction to computational topology the interested reader may consult Refs. [2, 5], for instance.

### 3.2.1 Alpha complexes

Let $X_\nu(t)$ be a point cloud as defined by Eq. (7). We construct persistent homology groups from a nested family of simplicial complexes. A *simplicial complex* $\mathcal{S}$ on $X_\nu(t)$ comprises the set $X_\nu(t)$ together with a collection $\mathcal{S}$ of subsets of $X_\nu(t)$. The defining property of a simplicial complex is that for all points $x \in X_\nu(t)$, the vertex $\{x\} \in \mathcal{S}$, and if $\tau \subseteq \sigma \in \mathcal{S}$, then $\tau \in \mathcal{S}$, i.e. $\mathcal{S}$ is closed under taking subsets. The elements of $\mathcal{S}$ are called its *simplices*. Combinatorially, this structure allows for the computation of various descriptors of its topology, in particular the homology groups of $\mathcal{S}$. We deliver details in Appendix A.1.

Let us construct the particular type of simplicial complexes employed in this work: alpha complexes. Clearly, for any three points in $X_\nu(t)$ that do not lie on a single straight line, a unique circumsphere passing through the points exists. Any two points can be trivially identified with a zero-dimensional circumsphere. We shall assume that the points in $X_\nu(t)$ are in general position. This excludes, for example, the possibility that three or more points are collinear or that four or more points lie on a single circle[3]. Then, any two or three points in $X_\nu(t)$ have a unique zero- or one-dimensional circumsphere passing through these points, respectively[4]. We call a circumsphere empty, if all points of $X_\nu(t)$ lie on or outside the sphere.

The *Delaunay complex*, $\mathrm{Del}(X_\nu(t))$, can be defined to consist of all points in $X_\nu(t)$ as well as those edges and triangles whose circumspheres are empty [47]. Speaking about terminology, a point is a zero-dimensional simplex, an edge between two points is a one-dimensional simplex and a triangle is a two-dimensional simplex. As described in Ref. [46], for point clouds in general position this procedure yields that the corresponding Delaunay complex is a simplicial complex, allowing for the construction of homology groups as described intuitively below.

The *Delaunay radius function* $\mathrm{Rad} : \mathrm{Del}(X) \to [0, \infty)$ is defined to map every simplex to the smallest radius of all its empty circumspheres. Intuitively, it provides a measure for the size of a simplex. In Fig. 4d the Delaunay complex of an example point cloud $X_\nu(t)$ as it appears in the Bose gas simulations is displayed for $\bar{\nu} = 0.6$. Note that simplices of different Delaunay radii are visually of distinct dominance, typically. Smaller simplices appear foremost around local accumulations of points, while simplices of larger radii mainly make up the large-scale structure between them.

Let $Qr \in [0, \infty)$ be some length scale. Capturing appearing structures of particular sizes, from the Delaunay radius function we finally construct *alpha complexes*[5] as its sublevel sets,

$$\alpha_r(X_\nu(t)) := \{\sigma \in \mathrm{Del}(X_\nu(t)) \,|\, \mathrm{Rad}(\sigma) \le Qr\}. \tag{10}$$

For all $0 \le r \le s$ we find $\alpha_r(X_\nu(t)) \subseteq \alpha_s(X_\nu(t))$. To this end, we obtain what is called a *filtration* of the Delaunay complex $\mathrm{Del}(X_\nu(t))$, that is, a nested sequence of alpha complexes little by little filling out all $\mathrm{Del}(X_\nu(t))$,

$$\emptyset \subseteq \alpha_{r_1}(X) \subseteq \cdots \subseteq \alpha_{r_\kappa}(X) = \mathrm{Del}(X), \tag{11}$$

with $r_i \le r_j$ for all $i < j$.

Again referring to the example point cloud $X_\nu(t)$, in Fig. 4 corresponding alpha complexes of different radii $Qr$ are displayed. Note that at a small radius such as $Qr = 1.0$ the alpha complex mainly reflects the local accumulations of points in $X_\nu(t)$. Topological structures such as holes are of tiny size and each connected component loosely corresponds to a local accumulation of points. Besides seemingly random connected structures, at intermediate radii comparably large-scale holes appear in the alpha complexes, such as visible in the

---

[3]While different definitions of general position exist across the literature, we employ the one used in Ref. [46].

[4]In general spatial dimension $d$ this would amount to any $2 \le j \le d + 1$ points $x_{i_1}, \ldots, x_{i_j}$ having a unique $(j-2)$-dimensional circumsphere passing through all these points.

[5]Generically, alpha complexes are simplicial subcomplexes of the Delaunay complex [5].

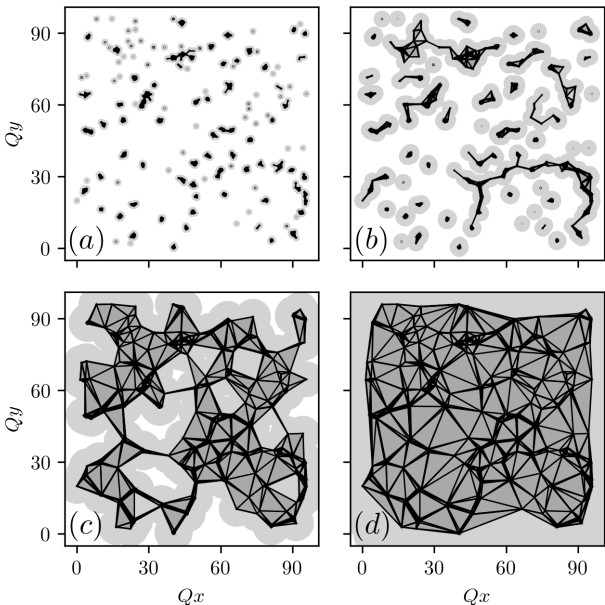

Figure 4: Alpha complexes of various radii $Qr$ of the point cloud $X_\nu(Qt = 3750)$ for $\bar{\nu} = 0.6$ as displayed in Fig. 3. Panel (a): $Qr = 1.0$. Panel (b): $Qr = 3.0$. Panel (c): $Qr = 7.0$. Panel (d): $Qr = 20.0$.

$Qr = 7.0$ alpha complex displayed in Fig. 4c. At even larger radii, the full Delaunay complex is recovered, in accordance with Eq. (11). Leading to the notion of persistent homology, it is a crucial insight that independent connected components disappear at a certain radius, merging with other components, and that holes only appear in alpha complexes of a certain radius and disappear again at a higher radius.

### 3.2.2 Persistent homology and the persistence diagram

This intuitive picture can be turned into a mathematical concept: persistent homology. In Appendix A.2, we provide a more rigorous introduction to it, while here we focus on capturing its intuitive essence.

Alpha complexes of zero radius only consist of the vertices, that is, all points contained in the point cloud $X_\nu(t)$. Certainly, the number of connected components in the alpha complex of zero radius equals the cardinality of $X_\nu(t)$. Increasing the radius, at a certain value a first edge between two vertices appears in the alpha complex. A previously independent connected component *dies*. We call the minimum radius at which it is not present anymore in the corresponding alpha complex its *death radius*. The radius rising further, more and more connected components die, merging into a larger and larger complex. From a certain radius onwards, only one connected component is present in the corresponding alpha complexes. In Fig. 4 the process of connected components merging one by one into larger complexes can be observed as the sequence of alpha complexes is traversed towards larger radii.

With radii increasing, in the sequence of alpha complexes holes begin to appear as is clearly visible in Figs. 4b and 4c. The minimum radius at which an independent hole first appears in the sequence of alpha complexes is called its *birth radius*. We say that it is *born* at its birth radius. Successively, a given hole is filled out with triangles in alpha complexes of rising radii, until from its *death radius* onwards the hole vanishes, being fully filled.

In fact, in simplicial homology independent connected components are described by *zero-dimensional homology classes* and independent holes by *one-dimensional homology classes*. If

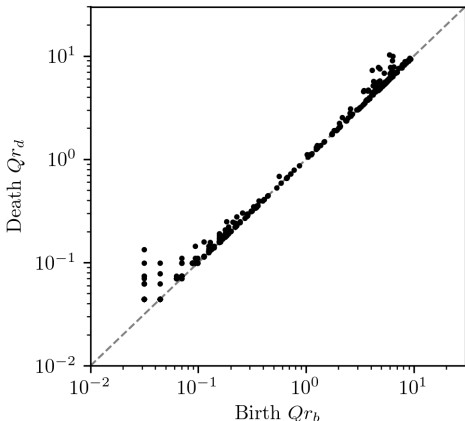

Figure 5: Persistence diagram of one-dimensional homology classes for the sequence of alpha complexes partially displayed in Fig. 4, $\mathrm{Dgm}_1(X_\nu(t))$.

the point clouds of interest lived in a higher-dimensional Euclidean space, one could continue analogously to describe the birth and death of higher-dimensional homology classes. This includes, for instance, independent enclosed voids represented by two-dimensional homology classes. Homology classes of dimension $\ell$, appearing and disappearing again as the sequence of alpha complexes is traversed, are collected in groups, the $\ell$-th *persistent homology groups*, cf. Appendix A.2.

Summarizing the structure of $\ell$-th persistent homology groups, the $\ell$-th *persistence diagram* $\mathrm{Dgm}_\ell(X_\nu(t))$, is defined to contain all birth radius-death radius pairs $(r_b, r_d)$ of $\ell$-dimensional homology classes appearing in the sequence of alpha complexes of $X_\nu(t)$, taking respective multiplicities into account for coinciding such pairs[6]. In Fig. 5 the persistence diagram of one-dimensional homology classes is displayed for the sequence of alpha complexes partially shown in Fig. 4. Certainly, in a persistence diagram all points lie above the diagonal $r_b = r_d$, since the death of any homology class happens at a higher radius than its birth. We find that in the bottom-left of the diagram an accumulation of pairs is present, corresponding to comparably small one-dimensional homology classes (holes). The partly vertical alignment of points can be attributed to the homogeneity of the square lattice, on which $X_\nu(t)$ resides. In addition, we find a second accumulation of pairs in the top-right of the diagram, corresponding to larger-size one-dimensional homology classes in corresponding alpha complexes. On these length scales birth and death radii are approximately independent from the microscopic lattice geometry.

### 3.2.3 Statistical measures: birth and death radii distributions

To obtain expectation values in the classical-statistical framework, ensemble-averages of quantities describing persistence diagrams of individual classical-statistical realizations are required. Persistence diagrams themselves are difficult objects to study statistically. Without modifications not even a statistical average can be defined unambiguously. Nevertheless, there exist multifarious quantities suitable for a statistical treatment [48]. We introduce two of these here, postponing the general description to Sec. 4.1. We explicitly construct classical-statistical ensemble-averages. To this end, let $X_\nu^{(i)}(t)$, $i \in \mathbb{N}$, be an ensemble of point clouds, all constructed from individual field realizations according to Eq. (7). Denote by $D_\ell^{(i)}(t) := \mathrm{Dgm}_\ell(X_\nu^{(i)}(t))$ the $\ell$-th persistence diagram of the $i$-th such point cloud. Let $\sigma > 0$

---

[6]The persistence diagram is a finite multiset of points in $\mathbb{R}^2$, also taking respective multiplicities into account.

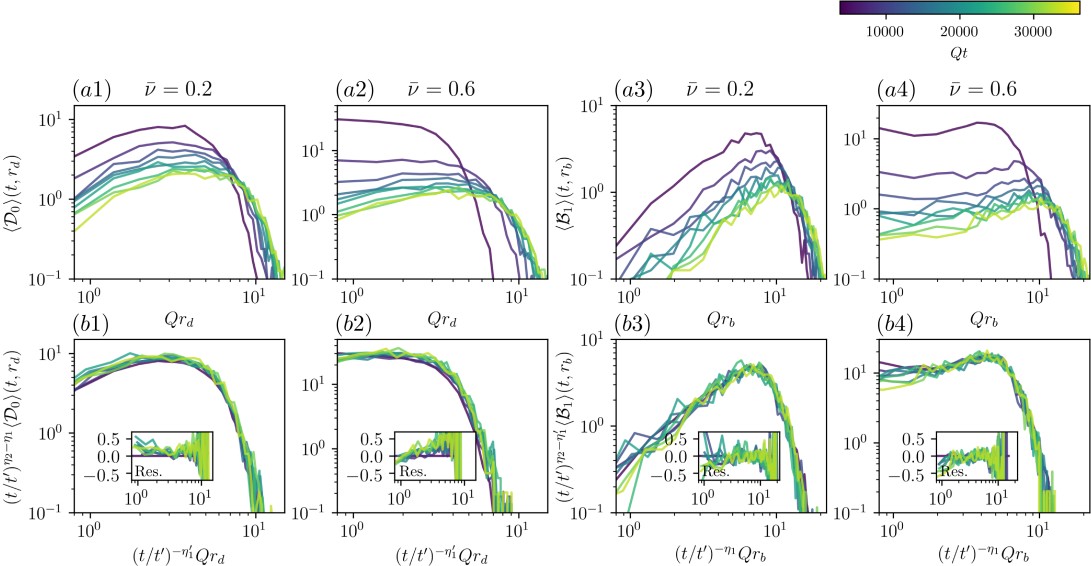

Figure 6: Birth and death radii distributions in the infrared. Columns 1 and 2: Death radii of of zero-dimensional homology classes. Columns 3 and 4: Birth radii of one-dimensional homology classes. Individual columns show data for the indicated filtration parameter, $\bar{\nu}$. Row 1: unrescaled distributions. Row 2: rescaled distributions. The employed time-dependent scaling exponents are displayed in Fig. 8.

be a constant. We define the expectation values of the *ℓ-th distribution of birth radii* and the *ℓ-th distribution of death radii* as

$$\langle \mathcal{B}_\ell \rangle (t, r_b) = \lim_{k \to \infty} \frac{1}{k} \sum_{i=1}^{k} \sum_{(r'_b, r'_d) \in D_\ell^{(i)}(t)} \frac{1}{2\pi\sigma^2} \exp\left( -\frac{(r_b - r'_b)^2}{2\sigma^2} \right), \qquad (12a)$$

$$\langle \mathcal{D}_\ell \rangle (t, r_d) = \lim_{k \to \infty} \frac{1}{k} \sum_{i=1}^{k} \sum_{(r'_b, r'_d) \in D_\ell^{(i)}(t)} \frac{1}{2\pi\sigma^2} \exp\left( -\frac{(r_d - r'_d)^2}{2\sigma^2} \right), \qquad (12b)$$

respectively. Note that these distributions are statistically well-behaved, such that averages and the denoted limits exist [49]. The parameter $\sigma$ is chosen sufficiently small, such that numerical outcomes are independent from its particular value.

## 3.3 Growing geometric structures in persistent homology

Using a computational topology pipeline as described in Appendix B, we can numerically investigate birth and death radii distributions for different filtration parameters $\bar{\nu}$ in the aforementioned Bose gas simulations. For large length scales, in Fig. 6 death radii distributions of zero-dimensional homology classes and birth radii distributions of one-dimensional homology classes are displayed at times between $Qt = 3750$ and $Qt = 35625$. Zero-dimensional persistent homology classes are always born at radius $Qr_b = 0$, turning the distribution of birth radii of zero-dimensional homology classes trivial. The occurring oscillations in distributions are due to statistical uncertainties, being computed from only a finite number of classical-statistical samples.

We first discuss unrescaled variants of the displayed distributions. It is important to note that in any of the distributions the maximum number of counts in birth and death radii distributions decreases with time. Simultaneously, the steep decline at largest radii in birth and

death distributions constantly shifts to higher radii. Clearly, these are manifestations of geometric structures in the system growing at large length scales as conjectured in Sec. 3.1 from the point clouds themselves. Beyond this, the approximately constant form of the distributions already provides a first hint at self-similar dynamics.

In first death radii distributions a clear peak is visible, in particular for $\bar{\nu} = 0.2$ as displayed in Fig. 6, panel (a3). Point clouds for small $\bar{\nu}$-values being dominated by accumulations of points around vortex nuclei, we expect this distinguished length scale to provide a measure for the average inter-vortex distance. At higher $\bar{\nu}$-values such as $\bar{\nu} = 0.6$ the peak is blurred by means of bulk points entering corresponding point clouds.

### 3.4 Unveiling a spectrum of scaling exponents

Motivated by the approximately constant form of the distributions displayed in Fig. 6, we examine whether they can be consistently described by a self-similar scaling ansatz. We say that birth and death radii distributions *scale self-similarly*, if exponents $\eta_1, \eta_1'$ and $\eta_2$ exist, such that for all times $t, t'$,

$$\langle \mathcal{B}_\ell \rangle (t, r_b) = (t/t')^{\eta_1' - \eta_2} \langle \mathcal{B}_\ell \rangle (t', (t/t')^{-\eta_1} r_b), \tag{13a}$$

$$\langle \mathcal{D}_\ell \rangle (t, r_d) = (t/t')^{\eta_1 - \eta_2} \langle \mathcal{D}_\ell \rangle (t', (t/t')^{-\eta_1'} r_d). \tag{13b}$$

In Sec. 4.3 we deduce this particular form of scaling behavior from a scaling ansatz to a more general quantity that describes persistent homology groups, the asymptotic persistence pair distribution. Notice that in this scaling ansatz a possible dependence on the dimension $\ell$ of homology classes is neglected, supported by numerics.

Using the numerical protocol described in Appendix G, scaling exponents are extracted from birth and death radii distributions of one-dimensional homology classes. Given a time $Qt_{\min}$, birth and death radii distributions at times $Qt_{\min}$, $Qt_{\min} + 625$ and $Qt_{\min} + 1250$ are fitted simultaneously against distributions at reference time $Qt' = 3750$. A measure for the quality of a self-similar description of the investigated distributions is provided by means of residuals. For instance, for the distribution of birth radii residuals at time $Qt$ are computed as

$$\text{Res.}(\langle \mathcal{B}_\ell \rangle)(t, r_b) := \frac{(t/t')^{\eta_1' - \eta_2} \langle \mathcal{B}_\ell \rangle (t', (t/t')^{-\eta_1} r_b)}{\langle \mathcal{B}_\ell \rangle (t, r_b)} - 1. \tag{14}$$

Indeed, distributions can be consistently rescaled by means of the scaling ansatz described in Eqs. (13a) and (13b). This can be deduced from Fig. 6 with residuals of rescaled distributions scattering approximately evenly around zero. Note that distributions of both zero- and one-dimensional homology classes can be consistently rescaled with the same triple of exponents, validating that in the scaling ansatz we neglected a possible $\ell$-dependence. However, *filtration parameter- and time-dependent* scaling exponents are necessary for a successful rescaling.

In Fig. 7 we show the scaling exponents for a single minimum fitting time $Qt_{\min}$, highlighting the size of error bars. Errors origin from a finite number of classical-statistical samples taken into account and from fitting uncertainties. For values of $\bar{\nu} \lesssim 0.4$ the displayed exponent values approximately lie around 0.2. A rise in values takes place for $\bar{\nu} \gtrsim 0.5$, up to a maximum value of approximately 0.8. Thus, we make the crucial observation that a *continuous spectrum* of scaling exponents exists, depending on the filtration parameter $\bar{\nu}$.

Within error bars $\eta_1$ equals $\eta_1'$ at all $\bar{\nu}$-values investigated here. This provides numerical evidence for that birth and death radii show the same dynamics at large length scales. In addition, for all $\bar{\nu}$-values analyzed $\eta_2/\eta_1 = 4$ within the indicated error bars. This relation results from the bounded packing of homology classes of a given size into the constant lattice volume, as shown in Sec. 4.3.2.

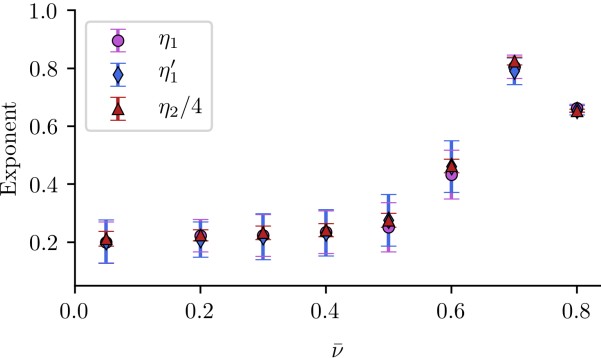

Figure 7: Persistent homology scaling exponents at $Qt_{\min} = 18750$.

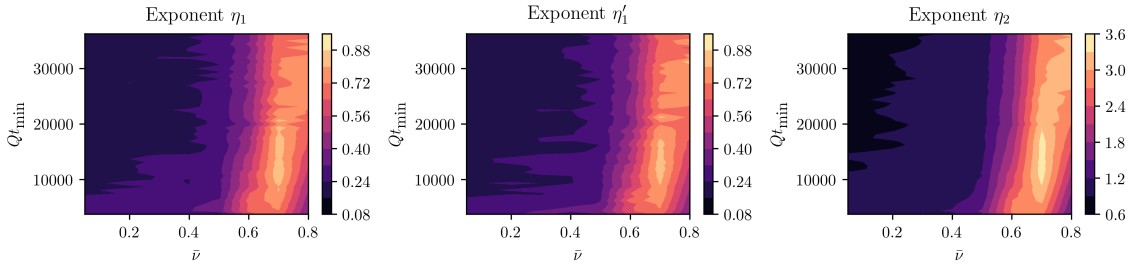

Figure 8: Persistent homology scaling exponents for different filtration parameters $\bar{\nu}$ and minimum fitting times $Qt_{\min}$.

Comprehensively, results are summarized in Fig. 8, in which exponents are displayed in the full $(\bar{\nu}, Qt_{\min})$-plane. The gradual shift of the peak in scaling exponents to higher $\bar{\nu}$-values with increasing fitting time $Qt_{\min}$ is a result of the redistribution of amplitude values with time, discussed in Sec. 5.1. The scattering of exponent values at larger $\bar{\nu}$-values is due to statistical uncertainties.

## 3.5 Scaling species and exponents mixing conjecture

An observation such as the existence of a whole spectrum of scaling exponents at large length scales requires an explanation. We conjecture that its appearance is linked to different dynamical scaling species occurring in the infrared of the two-dimensional Bose gas.

First, note that momenta in the infrared regime correspond to large length scales. Hence, if infrared dynamics is visible in quantities describing the persistent homology of alpha complexes, it will show at correspondingly large birth and death radii. Vice versa, if ultraviolet physics is visible in persistent homology, it will show up at comparably small birth and death radii. To this end, we identify the regime of large birth and death radii in their distributions with the infrared regime of the system. This offers the possibility of linking aforementioned results to known momentum space dynamics of physical quantities.

In addition, for positive scaling exponents $\eta_1 = \eta_1'$ and $\eta_2$ the scaling ansatz described by Eqs. (13a) and (13b) corresponds to a blow-up of length scales as a power-law with exponent $\eta_1$, as we detail in Sec. 4.3. Hence, a comparison of the exponent $\eta_1$ with scaling exponents appearing in power-laws of further physical length scales is reasonable.

We restrict the following discussion to $\eta_1$. For $\bar{\nu} \lesssim 0.4$, the exponent $\eta_1$ meets the value of 1/5 associated to the anomalous vortex kinetics nonthermal fixed point [37,42] and confirmed by the self-similar dynamics of occupation number spectra in the given simulations, Eq. (6).

Point clouds, alpha complexes as well as birth and death radii distributions reflect the occurring vortex dynamics for small $\bar{\nu}$, correspondingly. This is in accordance with the observation made in Sec. 3.1 that for $\bar{\nu} \lesssim 0.4$ point clouds mainly comprise accumulations of points around vortex nuclei.

The exponent $\eta_1$ increases with $\bar{\nu}$ up to maximum values of between 0.7 and 0.9 depending on $Qt_{\min}$, cf. Fig. 8 — a value which is significantly different from $1/5$. We take a small detour to provide a physical interpretation for this phenomenon.

Collectively, the vortices show anomalous kinetics and dominate point clouds at low $\bar{\nu}$-values: $\eta_1(\bar{\nu} = 0.05) \approx 0.2$. It is well-known, however, that the two-dimensional nonrelativistic Bose gas not only exhibits the anomalous vortex kinetics nonthermal fixed point with $\beta = 0.2$, but also incorporates strong wave turbulence characterized by $\beta = 0.5$ [32,37,42,50]. If the vortices were absent or coupled strongly to sound excitations in the bulk, only self-similar scaling with $\beta = 0.5$ would be visible, as argued for in Ref. [37]. Motivated by this, we infer that in the configurations investigated it is sound excitations in the bulk that reflect strong wave turbulence. Correspondingly, if bulk points enter point clouds, then birth and death radii distributions might show scaling behavior deviating from $\eta_1 = 0.2$. As can be seen in Figs. 3, 7 and 8 this is the case for growing $\bar{\nu}$-values and explains the increase of $\eta_1$. With this admittedly loose association of bulk points to strong wave turbulence and vortex nuclei points to anomalous vortex kinetics in mind, we refer to the underlying phenomenon as *scaling species mixing* in point clouds.

Yet, the maximum value of $\eta_1(\nu)$ exceeds 0.5 significantly for all $Qt_{\min}$. A heuristic geometric explanation proceeds as follows. Restrict to the dynamics of a single classical-statistical field configuration and corresponding point clouds $X_\nu(t)$. Let $Y_\nu(t) \subseteq X_\nu(t)$ be associated to anomalous vortex kinetics and $Z_\nu(t) \subseteq X_\nu(t)$ associated to strong wave turbulence in the bulk, such that $X_\nu(t) = Y_\nu(t) \cup Z_\nu(t)$. The alpha complexes of $X_\nu(t)$, $\alpha_r(X_\nu(t))$, however, do not simply decay into $\alpha_r(Y_\nu(t))$ and $\alpha_r(Z_\nu(t))$. Instead, depending on the precise arrangements of points in $Y_\nu(t)$ and $Z_\nu(t)$, there may be a lot of simplices contained in $\alpha_r(X_\nu(t))$ which incorporate points of both $Y_\nu(t)$ and $Z_\nu(t)$. In addition, simplices that only consist of points in $Y_\nu(t)$ or $Z_\nu(t)$ can be very different from the ones in $\alpha_r(Y_\nu(t))$ and $\alpha_r(Z_\nu(t))$. The construction of alpha complexes from $Y_\nu(t)$ and $Z_\nu(t)$ is a highly nonlinear process. Birth and death radii distributions can reflect this behavior.

# 4 Persistent homology observables and self-similarity

In this section we embed alpha complexes and persistent homology descriptors into the classical-statistical regime of quantum field theory (QFT). By means of functional summaries of persistence diagrams, this leads to the definition of persistent homology observables. In quite a few examples of these the same integral kernel appears, which we call the asymptotic persistence pair distribution. This paves the way to a self-similar scaling approach for the asymptotic persistence pair distribution, whose outgrowths for birth and death radii distributions are given by Eqs. (13a) and (13b). In Sec. 3.4 this particular scaling behavior has been shown to describe simulation outcomes well.

## 4.1 Persistent homology observables via functional summaries

Naturally, studying persistent homology in QFT requires a statistical treatment. Persistence diagrams themselves, however, do not admit a clear notion of averages [48]. Instead, we propose to focus on so-called functional summaries, providing general statistically well-behaved descriptors of persistence diagrams. In Sec. 4.2 we reveal that the investigated birth and death radii distributions given by Eqs. (12a) and (12b) are corresponding examples.

Let $\mathscr{D}$ be the space of persistence diagrams, that is, the space of finite multisets of points within $\{(r_b, r_d) \in [0, \infty)^2 \,|\, r_d \geq r_b\}$. Let $\mathscr{F}$ be a collection of functions, $f : \Omega \to \mathbb{R}$ for all $f \in \mathscr{F}$, $\Omega$ being a compact space. Following Ref. [49], a *functional summary* is in full generality any map from the space of persistence diagrams to a collection of functions, $F : \mathscr{D} \to \mathscr{F}$.

Upon the classical-statistical approximation, expectation values of quantum observables are computed as ensemble-averages of classical field configurations, which are time-evolved via the corresponding classical equation of motion starting from fluctuating initial conditions. The range of validity of this approximation is typically restricted to high occupation numbers [32]. We propose to proceed analogously for functional summaries of persistence diagrams. To this end, any such summary $F$ may be evaluated on the level of individual field configurations and its expectation value $\langle F \rangle$ computed as the ensemble-average. We assume that the range of validity of this approach coincides with the well-known classical-statistical regime. Certainly, for any functional summary $F$ this proposal requires the existence of a corresponding linear operator $\mathcal{F}$, such that in the classical-statistical regime for any $s \in \Omega$,

$$\mathrm{tr}(\rho(t)\mathcal{F})(s) = \langle F \rangle(t, s), \tag{15}$$

$\rho(t)$ being the time-dependent density operator of interest, the trace taken over the corresponding quantum theory Hilbert space and the right-hand side being computed via the aforementioned evaluation scheme. However, the existence of such an operator $\mathcal{F}$ is a priori not clear and will be discussed in a future work.

We need to assure that in the limit of averaging infinitely many individual functional summaries of field configurations the statistical mean of the functional summary is recovered. This is guaranteed for by a mathematical statement on the pointwise convergence of so-called equicontinuous and uniformly bounded functional summaries, the details of which can be found in Proposition 1 of Ref. [49]. For the sake of this statement we restrict our proposal to functional summaries of persistence diagrams with these two fairly general conditions. By means of the described classical-statistical evaluation scheme we refer to such functional summaries as *persistent homology observables*.

We want to stress that this proposal is neither restricted to the computation of persistent homology from equal-time alpha complexes, that is, alpha complexes computed from point clouds constructed at individual instances of time as done in this work, nor to alpha complexes themselves.

## 4.2 The asymptotic persistence pair distribution and geometric quantities

Let $F : \mathscr{D} \to \mathscr{F}$ be a functional summary in the above sense. We say that $F$ is *additive*, if $F(D + E) = F(D) + F(E)$ for any two persistence diagrams $D, E \in \mathscr{D}$. Here, $D + E$ denotes the sum of multisets, that is, the union of $D$ and $E$ with multiplicities of elements in both $D$ and $E$ added.

Let $D(t) \in \mathscr{D}$ be a persistence diagram computed at time $t$ as specified in Sec. 3.2.2 and $F$ an additive functional summary. We then find for all $s \in \Omega$,

$$\begin{aligned}
F(D(t))(s) &= \sum_{(r_b, r_d) \in D(t)} F(\{(r_b, r_d)\})(s) \\
&= \int_0^\infty dr_b' \int_0^\infty dr_d' \, F(\{(r_b', r_d')\})(s) \, \mathfrak{P}(t, r_b', r_d'),
\end{aligned} \tag{16}$$

with the *persistence pair distribution*

$$\mathfrak{P}(t, r_b', r_d') := \sum_{(r_b, r_d) \in D(t)} \delta(r_b' - r_b) \delta(r_d' - r_d), \tag{17}$$

$\delta$ denoting the Dirac delta function.

Let $(D_\ell^{(i)}(t))_{i\in\mathbb{N}} \subset \mathscr{D}$ be a classical-statistical ensemble of persistence diagrams describing $\ell$-dimensional persistent homology classes at time $t$. We denote the persistence pair distribution of $D_\ell^{(i)}(t)$ by $\mathfrak{P}_\ell^{(i)}(t)$ and define the *asymptotic persistence pair distribution*, $\langle\mathfrak{P}_\ell\rangle$, at any time $t$ implicitly, requiring that for any equicontinuous and uniformly bounded functional summary $F$ as in the above proposal,

$$
\int_0^\infty dr_b' \int_0^\infty dr_d'\, F(\{(r_b', r_d')\})(s)\, \langle\mathfrak{P}_\ell\rangle(t, r_b', r_d')
$$
$$
:= \lim_{k\to\infty} \frac{1}{k} \sum_{i=1}^k \int_0^\infty dr_b' \int_0^\infty dr_d'\, F(\{(r_b', r_d')\})(s)\, \mathfrak{P}_\ell^{(i)}(t, r_b', r_d'), \qquad (18)
$$

for arbitrary $s \in \Omega$.

Functional summaries of relevance in this work include the distribution of birth and death radii that have been defined in Eqs. (12a) and (12b), respectively. With an obstacle to be described below, both can be computed as marginal distributions of $\langle\mathfrak{P}_\ell\rangle$,

$$
\langle\mathcal{B}_\ell\rangle(t, r_b) = \int_0^\infty dr_d\, \langle\mathfrak{P}_\ell\rangle(t, r_b, r_d), \qquad (19a)
$$

$$
\langle\mathcal{D}_\ell\rangle(t, r_d) = \int_0^\infty dr_b\, \langle\mathfrak{P}_\ell\rangle(t, r_b, r_d). \qquad (19b)
$$

In addition, we define the persistence distribution, that is, the distribution of $r_d - r_b$,

$$
\langle\mathcal{P}_\ell\rangle(t, r) = \int_0^\infty dr_d\, \langle\mathfrak{P}_\ell\rangle(t, r_d - r, r_d). \qquad (20)
$$

Natural quantities to study are the $\ell$-th Betti numbers $\langle\beta_\ell\rangle(t, r)$. Intuitively, the zeroth Betti number $\langle\beta_0\rangle(t, r)$ specifies the number of connected components minus one[7] present in the alpha complex of radius $Qr$ and the first Betti number $\langle\beta_1\rangle(t, r)$ specifies the corresponding number of holes. Being zero in the present work, higher Betti numbers count how many nontrivial higher-dimensional homology classes are present in corresponding complexes. Betti numbers can be computed from the asymptotic persistence pair distribution via

$$
\langle\beta_\ell\rangle(t, r) = \int_0^r dr_b \int_r^\infty dr_d\, \langle\mathfrak{P}_\ell\rangle(t, r_b, r_d). \qquad (21)
$$

A mathematical obstacle appears with regard to definitions such as Eqs. (19a) and (19b). A priori, the sets of functions $\langle\mathcal{B}_\ell\rangle(t, r_b)$, of $\langle\mathcal{D}_\ell\rangle(t, r_d)$, of $\langle\mathcal{P}_\ell\rangle(t, r)$ and of $\langle\beta_\ell\rangle(t, r)$ are not equicontinuous. However, only functional summaries which have this property are persistent homology observables in the sense of Sec. 4.1. For all positive $\sigma$ we define

$$
\zeta_\sigma(s) := \frac{1}{\sqrt{2\pi\sigma^2}} \exp\left(-\frac{s^2}{2\sigma^2}\right). \qquad (22)
$$

By convolution with it at each time individually, sets of functions such as $\langle\mathcal{B}_\ell\rangle(t, r_b)$ can be rendered equicontinuous[8]. In fact, this way Eqs. (12a) and (12b) for birth and death radii distributions arise from Eqs. (19a) and (19b). In everything that follows we omit the convolution

---

[7]We work with reduced homology groups. Thus, the zeroth Betti number actually counts the number of connected components minus one.

[8]Indeed, for any $\sigma > 0$ a constant $C_\sigma > 0$ exists, such that for all possible functions $\langle\mathcal{B}_\ell\rangle(t, r_b)$, $\partial(\langle\mathcal{B}_\ell\rangle * \zeta_\sigma)(t, r)/\partial r = (\langle\mathcal{B}_\ell\rangle * \zeta_\sigma')(t, r) < C_\sigma$, the prime indicating taking the first derivative. Here we employed that in the lattice framework all functions such as $\langle\mathcal{B}_\ell\rangle(t, r_b)$ are uniformly bounded.

procedure in notations. As mentioned previously, the convolution procedure is numerically irrelevant. In computations, convergence of persistent homology observables is numerically verified, cf. Appendix F.

The average number of persistent homology classes is encoded in $\langle \mathfrak{P}_\ell \rangle$, too,

$$\langle n_\ell \rangle(t) = \int_0^\infty dr_b \int_0^\infty dr_d \, \langle \mathfrak{P}_\ell \rangle(t, r_b, r_d). \tag{23}$$

Various length scales may be constructed from $\langle \mathfrak{P}_\ell \rangle$. An interesting length scale is the average maximum death radius $\langle r_{d,\ell,\max} \rangle(t)$, which can be computed from the asymptotic persistence pair distribution via[9]

$$\langle r_{d,\ell,\max} \rangle(t) = \lim_{p \to \infty} \left( \int_0^\infty dr_b \int_0^\infty dr_d \, r_d^p \, \langle \mathfrak{P}_\ell \rangle(t, r_b, r_d) \right)^{1/p}. \tag{24}$$

Analogously, the average maximum birth radius can be computed. The average number of persistent homology classes and the average maximum death (birth) radius constitute persistent homology observables as constructed above.

## 4.3 Self-similar scaling approach

By means of the scaling behavior visible in birth and death radii distributions, in Sec. 3.4 we have already begun the study of self-similarity in persistent homology observables in the vicinity of a nonthermal fixed. Here, we introduce a more general scaling ansatz for the asymptotic persistence pair distribution. We provide a heuristic packing argument relating the appearing scaling exponents.

In Appendix D we provide a brief discussion on the relation between the self-similar scaling ansatz described here and known notions of self-similar scaling appearing across the literature.

### 4.3.1 Scaling ansatz to the asymptotic persistence pair distribution

Let $\langle \mathfrak{P}_\ell \rangle(t, r_b, r_d)$ be a time-dependent asymptotic persistence pair distribution as it appears in Eq. (18). We say that $\langle \mathfrak{P}_\ell \rangle(t, r_b, r_d)$ scales self-similarly, if exponents $\eta_1, \eta_1'$ and $\eta_2$ exist, such that for all times $t, t'$,

$$\langle \mathfrak{P}_\ell \rangle(t, r_b, r_d) = (t/t')^{-\eta_2} \langle \mathfrak{P}_\ell \rangle(t', (t/t')^{-\eta_1} r_b, (t/t')^{-\eta_1'} r_d). \tag{25}$$

Due to the time-dependence of $\langle \mathfrak{P}_\ell \rangle$ derived geometric quantities become time-dependent, too. Immediately, from Eq. (25) for birth and death radii distributions the scaling behavior described by Eqs. (13a) and (13b) follows. Assuming $\eta_1 = \eta_1'$, the persistence distribution scales as

$$\langle \mathcal{P}_\ell \rangle(t, r) = (t/t')^{\eta_1 - \eta_2} \langle \mathcal{P}_\ell \rangle(t', (t/t')^{-\eta_1} r). \tag{26}$$

The total number of persistence pairs scales as

$$\langle n_\ell \rangle(t) = (t/t')^{\eta_1 + \eta_1' - \eta_2} \langle n_\ell \rangle(t') \tag{27}$$

and the average maximum death radius as

$$\langle r_{d,\ell,\max} \rangle(t) = (t/t')^{\eta_1} \langle r_{d,\ell,\max} \rangle(t'). \tag{28}$$

---

[9]Given positive real numbers $y_1, \ldots, y_m$, one obtains their maximum via $\max\{y_1, \ldots, y_m\} = \lim_{p \to \infty} (\sum_{i=1}^m y_i^p)^{1/p}$. From this, the given formula derives.

Though not explicitly given here, the average maximum birth radius scales the same way. This provides evidence for the geometric intuition of persistence length scales blowing up or shrinking in the course of time upon self-similar scaling.

Provided that $\eta_1 = \eta_1'$, the $\ell$-th Betti numbers scale as

$$\langle \beta_\ell \rangle (t, r) = (t/t')^{2\eta_1 - \eta_2} \langle \beta_\ell (t', (t/t')^{-\eta_1} r). \tag{29}$$

### 4.3.2 A heuristic packing relation

We assume that $\eta_1 = \eta_1'$ and consider a general spatial dimension $d$ here. A fairly general heuristic argument leads to the packing relation $\eta_2 = (2 + d)\eta_1$. Intuitively, the argument encodes that only a finite number of persistent homology classes of a given size can be packed into a constant volume $V$.

Let point clouds be dominated by a time-dependent length scale $L(t)$. The $d$-dimensional volume $V$ in which the point clouds reside is kept constant. Heuristically, a number $\langle n_{d-1} \rangle (t)$ of $(d-1)$-dimensional persistent homology classes fits into $V$, with this number scaling as

$$\langle n_{d-1} \rangle (t) \sim \frac{V}{L(t)^d}, \tag{30}$$

since the volume that each $(d-1)$-dimensional persistent homology class occupies generically may scale as $\sim L(t)^d$. Inferring the scaling of length scales as described by Eq. (28), that is, $L(t) \sim t^{\eta_1}$, we find

$$\langle n_{d-1} \rangle (t) \sim t^{-d\eta_1}. \tag{31}$$

On the other hand, from Eq. (27) we obtain

$$\langle n_{d-1} \rangle (t) \sim t^{2\eta_1 - \eta_2}. \tag{32}$$

Hence,

$$\eta_2 = (2 + d)\eta_1, \tag{33}$$

which shows that persistent homology observables represent in a direct fashion the geometry at hand.

Of course, the assignment of occupied volumes to $(d-1)$-dimensional homology classes is highly heuristic, bearing in mind that a homology class is an equivalence class of many cycles within a simplicial complex, rendering any such mapping ambiguous. However, one may use elements of the proof of the Wasserstein stability theorem for persistence diagrams, carried out in Ref. [44], to deduce Eq. (33) more rigorously from physically reasonable assumptions. In Appendix C we sketch the corresponding derivation, provided in detail in Ref. [51].

## 5 Exponent shifts, persistences and Betti number distributions

In this section the due explanation of temporal shifts of the scaling exponent spectrum observed in Sec. 3.4 is given as well as numerical outcomes for persistence distributions and Betti numbers. The latter provide further evidence for the suitability of the self-similar scaling ansatz for the asymptotic persistence pair distribution, as given by Eq. (25).

### 5.1 Amplitude redistribution-induced exponents shifts

The scaling exponents displayed in Fig. 8 change in time for $\bar{\nu} \gtrsim 0.5$. To discuss the origins of this effect, in Fig. 9 amplitude distributions are displayed for different times between $Qt = 3750$ and $Qt = 37500$. As is clearly visible, amplitudes redistribute with growing times

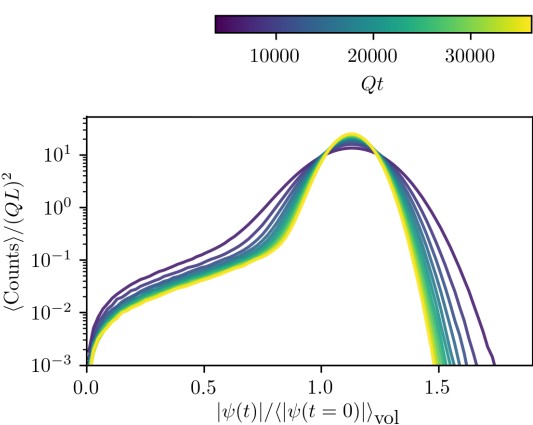

Figure 9: Distribution of amplitude-values at different times, averages taken across classical-statistical sampling runs.

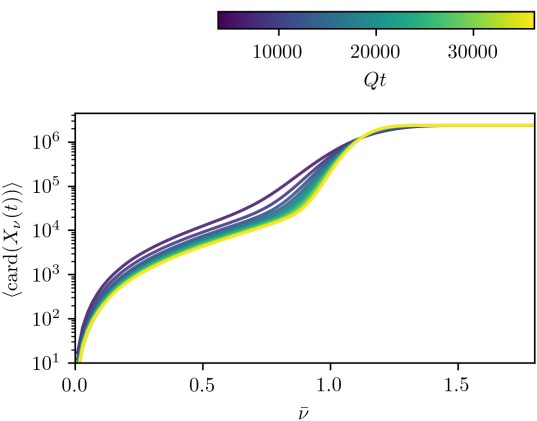

Figure 10: The average cardinality of point clouds varying with $\bar{\nu}$ at different times, averages taken across classical-statistical sampling runs.

towards the peak at around $|\psi(t)|/\langle|\psi(t=0)|\rangle_{\mathrm{vol}} \approx 1.05$. As indicated in Fig. 10, point clouds $X_\nu(t)$ with $\bar{\nu} \lesssim 1.0$ become sparser with time, that is, for a fixed $\bar{\nu}$ the cardinality of point clouds decreases.

As deduced earlier, at low $\bar{\nu}$-values point clouds are dominated by accumulations of points around vortex nuclei, while for $\bar{\nu} \gtrsim 0.4$ points in the bulk enter point clouds. With point clouds getting sparser in the course of time it is first bulk points to disappear from point clouds. Accumulations of points around vortex nuclei remain, as can be seen from Fig. 11, in which point clouds are displayed for different filtration parameters and times. Given the example point cloud for $\bar{\nu} = 0.5$ at time $Qt = 3750$, we observe that it is made up from accumulations of points (around vertices) mixed with random points in between, while at time $Qt = 11250$ the point cloud consists of nothing but the accumulations. The behavior of point clouds at $\bar{\nu} = 0.6$ is similar, although the point cloud at $Qt = 11250$ still contains random points associated to sound excitations between accumulations. Point clouds at $\bar{\nu} = 0.70$ only get sparser but still contain many bulk points.

The average maximum death radius of 1-dimensional persistent homology classes, $\langle r_{d,1,\max}\rangle(t)$, is displayed for different $\bar{\nu}$-values in Fig. 12. Comparably large fluctuations and outliners occur, since $\langle r_{d,1,\max}\rangle(t)$ is very sensitive to particular geometric arrangements

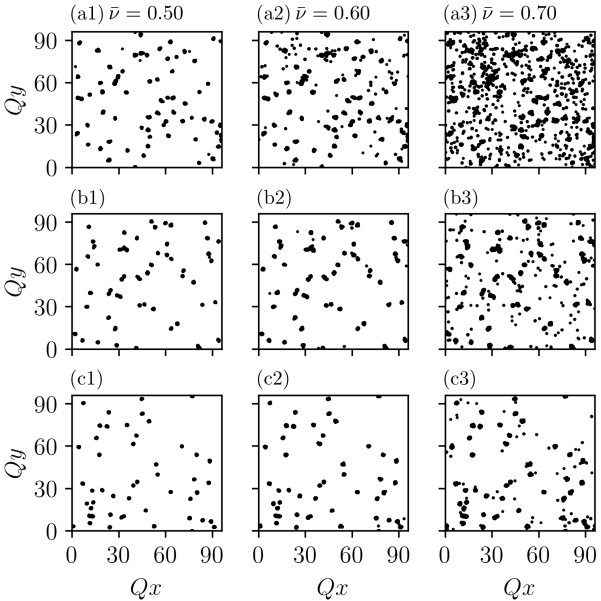

Figure 11: Example point clouds $X_\nu(t)$ for different $\bar{\nu}$-values as indicated. Row (a): time $Qt = 3750$. Row (b): $Qt = 7500$. Row (c): $Qt = 11250$.

of points in point clouds of individual classical-statistical samples. According to Eq. (28), if the system's asymptotic persistence pair distribution scales self-similarly in time and $\eta_1 = \eta_1'$, then $\langle r_{d,1,\max} \rangle(t) \sim t^{\eta_1}$. Indeed, $\langle r_{d,1,\max} \rangle(t)$ shows power-law behavior within individual periods of time and confirms the shifts in scaling exponents as indicated by the results displayed in Fig. 8, which have been deduced from birth and death radii distributions. For instance, for $\bar{\nu} = 0.6$ a shift occurs between times $Qt \approx 9000$ and $Qt \approx 13000$.

Recently, the phenomenon of prescaling has been discovered, that is, the rapid establishment of a universal scaling form of distributions long before the universal values of corresponding scaling exponents are realized [52, 53]. Although we also study time-dependent scaling exponents of constant-form distributions, we want to stress that in our case this is not a manifestation of prescaling. Instead, it is an artifact of the sharp cutoff at the filtration parameter to generate point clouds, rendering point clouds themselves and their persistent homology

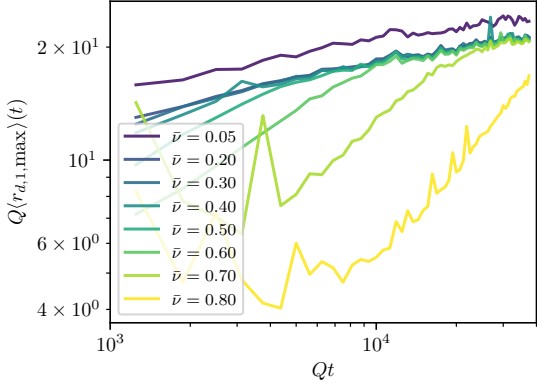

Figure 12: The average maximum death radius of 1-dimensional persistent homology classes varying with time, displayed for $\bar{\nu}$-values as indicated.

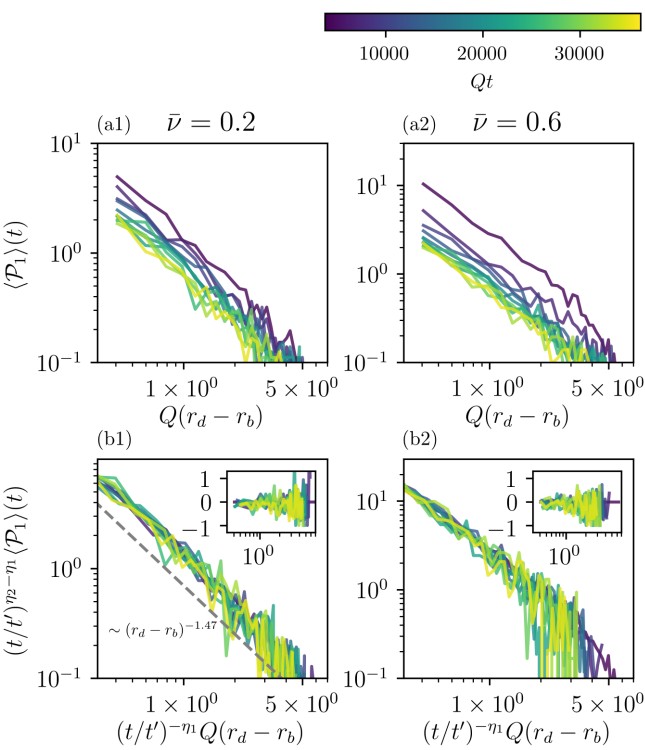

Figure 13: Persistence distributions. Each column shows data for the indicated filtration parameter, $\bar{\nu}$. The employed time-dependent scaling exponents are displayed in Fig. 8. Insets show corresponding residuals.

groups sensitive to amplitude redistribution effects.

## 5.2 Persistence distributions

In Fig. 13 persistence distributions for different filtration parameters are displayed. Again, fluctuations are due to statistical uncertainties. Distributions can be rescaled using time-dependent scaling exponents as given in Fig. 8. To this end, we attribute the observed behavior to the physics at large length scales. We want to emphasize that the persistence distributions at a low filtration parameter such as $\bar{\nu} = 0.2$ show distinctly a power-law behavior at all times. A power-law fit of the rescaled distributions for $\bar{\nu} = 0.2$ reveals a scaling with persistence as $\sim (r_d - r_b)^{-\zeta}$ with[10]

$$\zeta = 1.468 \pm 0.021. \tag{34}$$

The relation of the exponent $\zeta$ to known signatures of for example strong wave turbulence is to date not clear to us.

## 5.3 Betti numbers as a consistency check

In Sec. 4.3 we derived that if the asymptotic persistence pair distribution scales self-similarly, then Betti number distributions do so as well, described by Eq. (29). Having extracted scaling

---

[10]The power-law fit is first carried out for persistence values between $Q(r_d - r_b)_{\min} = 0.3125$ and $Q(r_d - r_b)_{\max} = 5.0$ at each of the times $Qt_i = 3750, 4375, \dots, 37500$, individually, to obtain values for $\zeta(t_i)$ and its fitting error at time $t_i$, $\Delta\zeta(t_i)$, $i = 1, \dots, N_i$. Subsequently, the value of $\zeta$ is defined to be the average of the obtained exponents. Its error squared, $\Delta\zeta^2$, is computed by means of standard error propagation as the sum of the temporal error squared and the sum of all $\Delta\zeta(t_i)^2/N_i^2$.

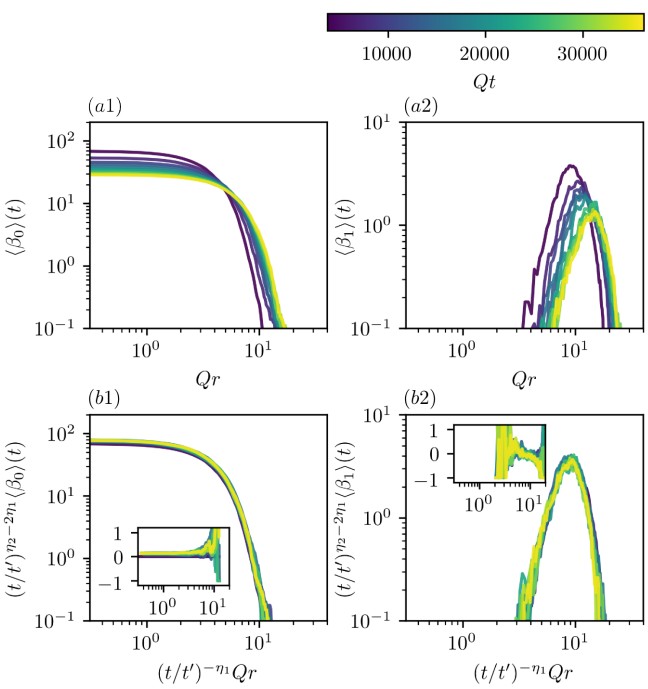

Figure 14: Betti number distributions for $\bar{\nu} = 0.2$ are shown for dimensions $\ell$ as indicated. The employed time-dependent scaling exponents are displayed in Fig. 8, setting $\eta'_1 := \eta_1$. Insets show corresponding residuals.

exponents from birth and death radii distributions in Sec. 3.4, we investigate Betti number distributions as a consistency check.

In Fig. 14 Betti number distributions for both zero- and one-dimensional homology classes are displayed at $\bar{\nu} = 0.2$. For all times $\langle \beta_0 \rangle(t, r)$ is a monotonically decreasing function, since zero-dimensional persistent homology classes are born at zero radius and $\langle \beta_0 \rangle(t, r)$ captures only their death. We find a peak in unrescaled $\langle \beta_1 \rangle(t, r)$, which, again, decreases in magnitude and shifts to higher radii as an indication of growing geometric structures.

Approximately, Betti numbers display self-similar scaling behavior. However, residuals of the rescaled $\langle \beta_0 \rangle(t)$ increase at large radii and $\langle \beta_1 \rangle(t)$ shows comparably large fluctuations. Nonetheless, rescaled Betti number distributions confirm previously extracted exponents.

# 6  Conclusions

In the present study we proposed a novel class of observables, persistent homology observables, to study the dynamical behavior of quantum fields. Serving as a prototype application, we investigated the self-similar dynamics at nonthermal fixed points in the classical-statistical approximation. Accompanied by mathematical considerations that guarantee, for example, for the convergence of averages, we studied functional summaries of persistent homology groups. We found that the notion of an asymptotic persistence pair distribution is a suitable probability measure for a self-similar scaling ansatz.

By means of simulations of the two-dimensional nonrelativistic Bose gas we revealed that the self-similar scaling dynamics characterizing nonthermal fixed points is a phenomenon that also appears in persistent homology observables. Crucially, this way we discovered a continuous spectrum of scaling exponents, depending on a filtration parameter that appears in the construction of point clouds. We provided a possible explanation in terms of scaling species

mixing associated to two different dynamical processes: Strong wave turbulence and anomalous vortex kinetics.

For all times investigated we found a power-law in persistence, possibly providing a direct indication in persistent homology observables for the presence of a turbulent cascade. It is currently unclear to us how to relate the deduced persistence power-law exponent to known power-law exponents appearing in occupation number spectra, typically signaling strong wave turbulence or hinting at topological defect structures [28, 37, 38].

Describing the wrapping of finite-size homology classes into a finite volume, by means of a packing relation we argued that self-similarity in persistent homology observables reflects the geometry at hand. Further exploring the relation between such geometric effects and conserved quantities associated to transport processes at nonthermal fixed points would be interesting, but lies outside the scope of this work.

Of particular relevance in the proposed persistent homology ansatz is the filtration function to generate point clouds from individual field configurations. We showed that already a simple variant such as the amplitude of the complex-valued fields can give rise to interesting observations. It is a feature of our analysis that the information on phase windings around vortex nuclei is not necessary in order to show the existence of further dynamical components beyond vortices. Nonetheless, we want to stress that at this point of the analysis scheme an immense freedom of choice exists, rendering the persistent homology ansatz highly flexible.

Also without such a filtration procedure the proposed methods can be applied to for instance point vortex models. Surpassing the present work, one does in principle not need a lattice to construct persistent homology groups. Even for fields with an arbitrary smooth and triangulable manifold as their domain there exist multifarious ways to construct persistent homology groups [5].

Myriad of interesting further applications of persistent homology within QFT exist. With regard to the recent experimental progress in handling ultracold quantum gases to simulate quantum dynamics [22, 23, 31]: What can we learn from a thorough persistent homology analysis of experimental data, including the investigation of different filtration functions? Can relative homology groups give new geometrical insights into the relevant physical processes?

Certainly, paths to illuminate also include analytics. Inter alia, for different types of random fields statistical statements could be made [54], and by means of integral geometry techniques predictions for alpha complexes of a class of random point clouds have been derived [46]. Using similar methods, is it possible to obtain analytic predictions for alpha complexes and their persistent homology in the context of quantum fields and path integrals?

Given the present study, we believe to have found a promising machinery to understand emergent connectivity and clustering structures far from equilibrium beyond the language of correlation functions via geometry and topology, providing a first step on the route of introducing persistent homology observables to QFT.

# Acknowledgements

We thank H. Edelsbrunner, K. Ölsböck, M. Prüfer, R. Ott, L. Shen, A. Chatrchyan, T.V. Zache and A.P. Orioli for discussions and collaborations on related work. We acknowledge support by the Interdisciplinary Center for Scientific Computing (IWR) at Heidelberg University, where part of the numerical work has been carried out.

**Funding information** This work is part of and supported by the DFG Collaborative Research Center "SFB 1225 (ISOQUANT)", and supported by the Deutsche Forschungsgemeinschaft (DFG, German Research Foundation) under Germany's Excellence Strategy EXC 2181/1

- 390900948 (the Heidelberg STRUCTURES Excellence Cluster). A.W. acknowledges support by the Klaus Tschira Foundation, and by the National Science Foundation under Grant No. 1440140 and the Clay Foundation, while she was in residence at the Mathematical Sciences Research Institute in Berkeley, California.

# A   The mathematics of persistent homology

The first part of this appendix serves as an intuitive entry point to standard algebraic topology concepts of relevance in this work. In the second part we construct persistent homology groups more rigorously than in the main text, including structural aspects.

Physically speaking, in this appendix we assume that all quantities are dimensionless. To this end, no factors of $Q$ appear.

## A.1   Relevant notions from algebraic topology

We introduce the notions of a simplicial complex, of chain groups and the boundary operator in order to finally introduce standard homology groups. For a thorough introduction to algebraic topology the reader may consult, for instance, Ref. [45].

Let $K$ be a simplicial complex. An element $\sigma \in K$ is a simplex of dimension $\ell$, if $\text{card}(\sigma) = \ell + 1$. Letting $\tau \subseteq \sigma$, we call $\tau$ a face of $\sigma$, and, vice versa, $\sigma$ a coface of $\tau$. The orientation of an $\ell$-simplex $\sigma = \{v_0, \ldots, v_\ell\} \in K$, is an equivalence class of permutations of its vertices, $(v_0, \ldots, v_\ell) \sim (v_{\pi(0)}, \ldots, v_{\pi(\ell)})$ if $\text{sign}(\pi) = 1$. An oriented simplex is denoted by $[\sigma]$. Geometrically, a simplex can be realized as the convex hull of $\ell + 1$ affinely independent points in $\mathbb{R}^d$, $d \geq \ell$. To this end, simplices of low dimension can be thought of as vertices, edges, triangles or tetrahedra, respectively.

Subcomplexes of a simplicial complex are subsets $L \subseteq K$ that are simplicial complexes, too. A nested sequence of complexes, $\emptyset = K_0 \subseteq K_1 \subset \cdots \subseteq K_k = K$ is called a filtration of the complex $K$.

We call the free Abelian group on the set of oriented $\ell$-simplices of a simplicial complex $K$ the $\ell$-th chain group $C_\ell$, where $[\sigma] = -[\tau]$ if $\sigma = \tau$ and $\sigma$ and $\tau$ are oriented differently. An element $c \in C_\ell$ is an $\ell$-chain, $c = \sum_i m_i[\sigma_i]$ with $\sigma_i \in K$ and $m_i \in \mathbb{Z}$. We define the boundary operator $\partial_\ell : C_\ell \to C_{\ell-1}$ to be the linear map defined by its action on a simplex $\sigma = [v_0, \ldots, v_\ell] \in c$,

$$\partial_\ell \sigma = \sum_j (-1)^j [v_0, v_1, \ldots, \hat{v}_j, \ldots, v_\ell], \tag{A.1}$$

$\hat{v}_j$ indicating that $v_j$ is deleted from the denoted sequence. Intuitively, the boundary operator maps an $\ell$-chain to its boundary, validating its nomenclature. A key feature is that $\partial_\ell \circ \partial_{\ell+1} = 0$, i.e. the boundary of a boundary is empty. Therefore the boundary operator connects the chain groups into an exact sequence, the chain complex $C_*$,

$$\cdots \to C_{\ell+1} \xrightarrow{\partial_{\ell+1}} C_\ell \xrightarrow{\partial_\ell} C_{\ell-1} \to \ldots. \tag{A.2}$$

To this end, the boundary group $B_\ell := \text{im}\,\partial_{\ell+1}$ and the cycle group $Z_\ell := \ker \partial_\ell$ are nested, $B_\ell \subseteq Z_\ell \subseteq C_\ell$.

The $\ell$-th homology group is then defined as $H_\ell := Z_\ell / B_\ell$. Its elements are equivalence classes of homologous cycles. Defined over a ring $\mathbb{Z}$, homology groups are $\mathbb{Z}$-modules. However, if defined over a field such as $\mathbb{Z}_2$ as done in the main text, homology groups become vector spaces.

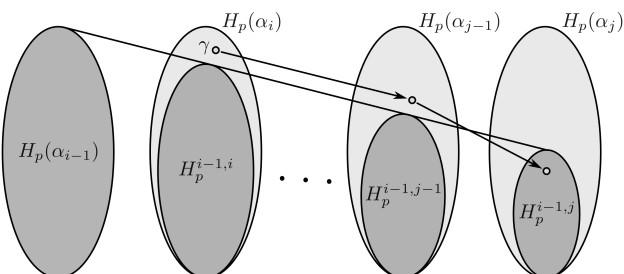

Figure 15: An illustration of the definitions of birth and death of homology classes. Picture inspired by Ref. [5].

## A.2 The construction and structure of persistent homology groups

We carry out the construction of persistent homology groups for the sequence of alpha complexes described in the main text, cf. Sec. 3.2.1. Let $X \subset \mathbb{R}^d$ be an arbitrary point cloud and $(\alpha_r(X))_{r \in [0,\infty)}$ its sequence of alpha complexes. The sequence is nested, $\alpha_r(X) \subseteq \alpha_s(X)$ for all $r \leq s$. $X$ being finite, only finitely many different $\alpha_r(X)$ exist, which can be specified by means of a finite set of different $r_i$, $i = 1, \dots, \kappa$. We abbreviate notations by means of $\alpha_i := \alpha_{r_i}(X)$ for all $i$.

For all $i \leq j$, the inclusion map $\iota^{i,j} : \alpha_i \to \alpha_j$ induces a homomorphism between homology groups, $\iota_\ell^{i,j} : H_\ell(\alpha_i) \to H_\ell(\alpha_j)$, for each dimension $\ell = 0, \dots, d$. To this end, the filtration of alpha complexes yields a sequence of homology groups,

$$0 \to H_\ell(\alpha_1) \to \cdots \to H_\ell(\alpha_\kappa) = H_\ell(\mathrm{Del}(X)). \tag{A.3}$$

Within this sequence, homology classes are born and later die again, when they become trivial or merge with other classes. With this intuition in mind, we set

$$H_\ell^{i,j} := \mathrm{im}(\iota_\ell^{i,j}), \qquad \forall\, 0 \leq i \leq j \leq \kappa, \tag{A.4}$$

as well as

$$\beta_\ell^{i,j} = \dim(H_\ell^{i,j}), \tag{A.5}$$

counting the number of homology classes that are born at or before $r_i$ and die after $r_j$.

To make the notions of birth and death of a simplex rigorous, let $\gamma \in H_\ell(\alpha_i)$. We say that $\gamma$ is born at $\alpha_i$ if $\gamma \notin H_\ell(\alpha_{i-1})$. If $\gamma$ is born at $\alpha_i$, then it dies entering $\alpha_j$, if it merges with an older class as going from $\alpha_{j-1}$ to $\alpha_j$, that is, $\iota_\ell^{i,j-1}(\gamma) \notin H_\ell^{i-1,j-1}$, but $\iota_\ell^{i,j}(\gamma) \in H_\ell^{i-1,j}$. The persistence of $\gamma$ is defined as $\mathrm{pers}(\gamma) := r_j - r_i$, if $\gamma$ is born at $\alpha_i$ and dies entering $\alpha_j$. For an illustration of this definition we refer to Fig. 15.

Actually, this intuitive definition has a conceptual drawback [2]. Any two homology classes that are born at the same birth radius $r_b$, one of them merging with the other one at a radius $r > r_b$, only die jointly at the death radius of the resulting homology class with highest death radius. A circumvention of this is provided by what is called the structure theorem of persistence modules [3, 4]. It states that up to isomorphism the family $((H_\ell(\alpha_i))_i, (\iota_\ell^{i,j})_{i \leq j})$ can be described by its persistence diagram as defined in the main text, cf. Sec. 3.2.2. An equivalent notion to the persistence diagram which regularly appears across topological data analysis literature is that of a barcode.

# B The computational pipeline

A variety of software exists designed to provide user-friendly and fast routines for the generation of simplicial complexes and the computation of persistent homology [2]. We employ the GUDHI library, which is a generic open source C++ library tailored to topological data analysis and higher dimensional geometry understanding [55]. In particular, with the simplex tree structure [56] it offers a handy data structure to store simplicial complexes. GUDHI employs the extensive CGAL library [57] to compute alpha complexes and uses a sophisticated algorithm to compute persistent homology groups. To give a rough indication of its speed, on a standard laptop alpha complexes of point clouds with approximately 100,000 data points can be analyzed in a few minutes, including the computation of persistent homology groups of all dimensions. For an overview of the computational cost of topological data analysis implementations across software solutions we refer to Ref. [2].

In this work we apply GUDHI functions to point clouds generated from individual field configurations according to Eq. (7). Obtaining persistent homology outcomes at various times for each field configuration, ensemble-averages are taken. Due to the lack of statistics, a direct analysis of the asymptotic persistence pair distribution $\langle \mathfrak{P}_\ell \rangle$ is unfeasible. Instead, for the $k = 72$ configurations investigated we have verified that the persistent homology observables $\langle \mathcal{B}_\ell \rangle(t, r_b)$, $\langle \mathcal{D}_\ell \rangle(t, r_d)$, $\langle \mathcal{P}_\ell \rangle(t, r)$ and $\langle \beta_\ell \rangle(t, r)$ converged properly. In Appendix F we analyze in detail the convergence behavior of persistent homology observables with $k$.

Of course, point clouds that are subsets of a regular lattice are generically not in general position, which can result in their Delaunay complexes not being simplicial complexes. GUDHI removes corresponding ambiguities by means of a built-in perturbation scheme for points out of general position. Effects of this procedure are not visible.

While simulations take periodic boundary conditions into account, alpha complexes of point clouds are computed non-periodically. Certainly, the toroidal topology of the lattice $\Lambda$ would have an effect on, for example, computed Betti numbers: The 2-torus has $\beta_0(T^2) = 0$, $\beta_1(T^2) = 2$ and $\beta_2(T^2) = 1$, which would at all times and radii add to $\langle \beta_\ell \rangle(t, r)$. The dynamics of point clouds and their persistent homology groups, however, would remain unaltered.

# C Packing relation from bounded total persistence

In Sec. 4.3.2 we provided a heuristic argument leading to the packing relation between scaling exponents in a self-similar scaling ansatz to the asymptotic persistence pair distribution,

$$\eta_2 = (2 + d)\eta_1. \tag{C.1}$$

Actually, under physically reasonable assumptions this relation can be properly derived. Here we outline this deduction. Details are provided in Ref. [51].

In Ref. [44] the notion of bounded total persistence has been introduced for the persistent homology of sublevel sets of a Lipschitz function $f : M \rightarrow \mathbb{R}$ with certain properties, $M$ being a connected, triangulable and compact metric space. For example, Lipschitz functions on the $d$-torus or the plane $[0, L]^d$, $L > 0$, have bounded total persistence. Given a point cloud $X \subset \mathbb{R}^d$ such as the $X_\nu(t)$ defined by Eq. (7), one can actually derive from the bounded total persistence an upper bound on the number of points in the persistence diagram of the sequence of alpha complexes. This upper bound scales with a particular length scale to the power of $-d$.

A statistical treatment of point clouds and persistence diagrams is necessary in order to define the asymptotic persistence pair distribution and the corresponding self-similar scaling ansatz. To this end, functional summaries as described in Sec. 4.1 play a key role. Properties

of point clouds, persistence diagrams and functional summaries such as self-averaging in the limit of large volumes can be turned rigorous.

Eventually, one can obtain Eq. (C.1) from the upper bound on the number of points in persistence diagrams. Central to the interpretation of Eq. (C.1) as describing the packing of homology classes into a constant volume is this upper bound.

## D  Relating persistent homology exponents to correlation function exponents

Typically, nonthermal fixed points and their properties are discussed in the framework of fixed-order correlation functions, both theoretically and experimentally [22, 23, 32, 58–60]. The self-similar scaling behavior at nonthermal fixed points allows for a grouping of far-from-equilibrium quantum systems into universality classes. Universality classes cover broad classes of far-from-equilibrium initial conditions, large ranges of relevant parameters and even theories with very different degrees of freedom [32]. Being a natural surrounding for universality, properties of nonthermal fixed points including scaling exponents have been derived within the renormalization group [61,62]. To this end, length scales derived from scaling correlation functions are expected to blow up or to shrink with a unique power-law in time.

If the asymptotic persistence pair distribution shows self-similar scaling as in Eq. (25), then any length scale derived from it scales in time as a power-law with exponent $\eta_1$, assuming $\eta_1 = \eta_1'$. As an example consider the average maximum death radius, defined in Eq. (24) and showing scaling as in Eq. (28). In light of this geometric analogy and the universality of scaling exponents at nonthermal fixed points, we expect that self-similar scaling behavior as extracted from correlation functions can typically be observed also in persistent homology observables.

## E  Details on the nonrelativistic Bose gas simulations

This appendix is devoted to provide details of the numerical setup to simulate the two-dimensional single-component nonrelativistic Bose gas in the classical-statistical regime. The computational implementation is described in Ref. [32].

Correspondingly, in the atomic gas let $a$ be the s-wave scattering length and $n$ its density. We define a diluteness parameter [32],

$$\zeta = \sqrt{na^3}, \tag{E.1}$$

and assume that $\zeta \ll 1$. A characteristic coherence length may be defined inversely via the momentum scale

$$Q = \sqrt{16\pi an}. \tag{E.2}$$

The average density, $n$, can be computed from the distribution function, $f(|\mathbf{p}|)$, $\mathbf{p}$ being the momentum, via

$$n = \int \frac{d^d p}{(2\pi)^d} f(|\mathbf{p}|). \tag{E.3}$$

For the validity of the classical-statistical approximation as well as extreme nonequilibrium conditions to trigger dynamics towards a nonthermal fixed point, we require a large characteristic mode occupancy, $f(Q) \gg 1$. Then, the dynamics becomes essentially classical and can

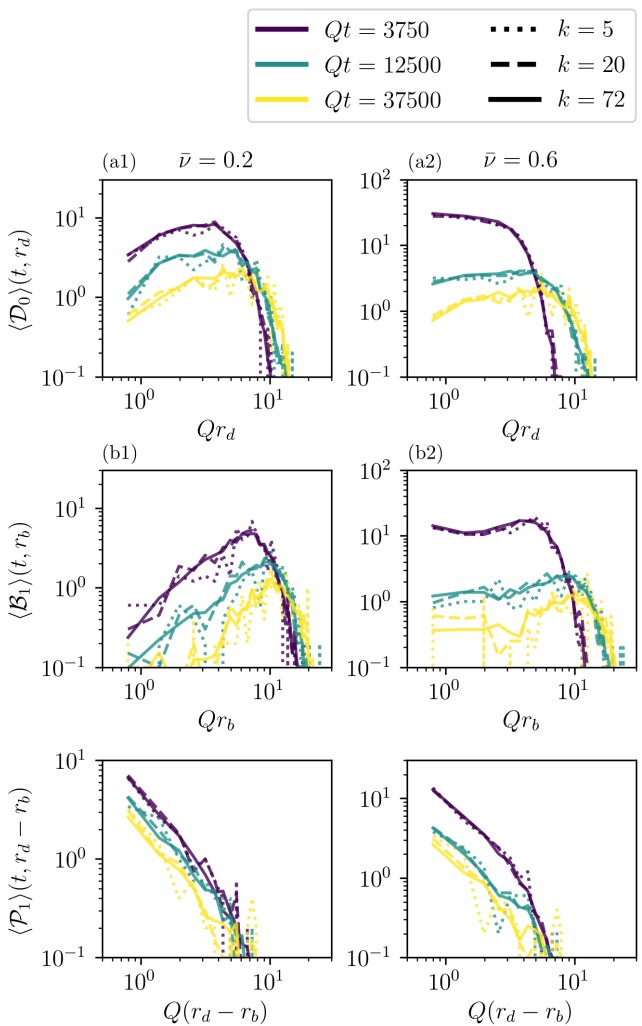

Figure 16: Birth and death radii distributions and persistence distributions in the infrared varying with time, displayed for $\bar{\nu}$-values and numbers of samples to average, $k$, as indicated.

be described by the time-dependent Gross-Pitaevskii equation for a nonrelativistic complex bosonic field, $\psi$,

$$i\partial_t \psi(t,\mathbf{x}) = \left( -\frac{\nabla^2}{2m} + g|\psi(t,\mathbf{x})|^2 \right) \psi(t,\mathbf{x}). \tag{E.4}$$

Fluctuating initial conditions, $f(\mathbf{p})$, are generated as samples of a Gaussian distibution with a width as described in Eq. (3). Each realization is evolved according to the discretized Gross-Pitaevskii equation, numerically solving the equation on a spatial lattice using a split-step method [32].

# F   Numerical convergence of persistent homology observables

In this appendix we provide results for how the different persistent homology observables of interest in the main text converge with the number of classical-statistical samples, $k$, increasing.

In Fig. 16 we display birth and death radii distributions as well as persistence distributions

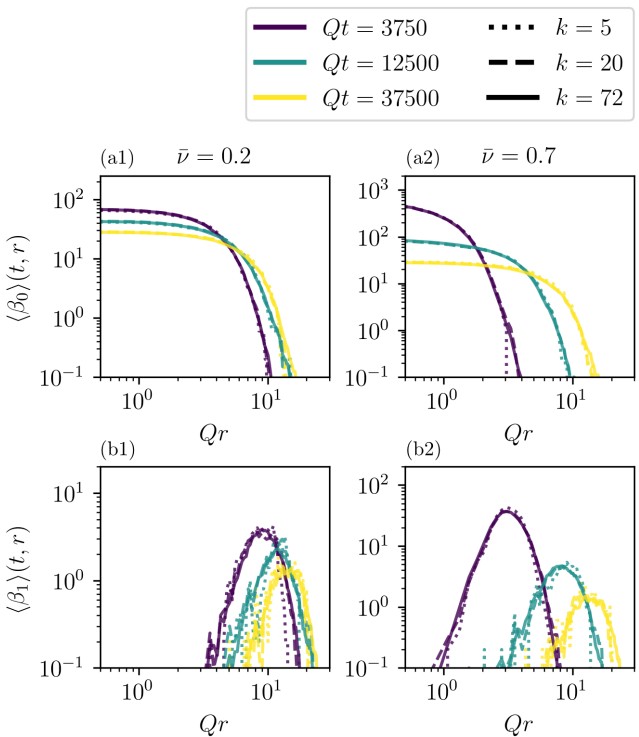

Figure 17: Betti number distributions in the infrared varying with time, displayed for $\bar{\nu}$-values and numbers of samples to average, $k$, as indicated.

for two values of $\bar{\nu}$, at different times within the persistent homology observables' self-similar scaling regime and for different values of $k$. It is clearly visible that occurring fluctuations decrease with $k$ increasing.

In Fig. 17 we display Betti numbers. In particular $\langle\beta_0\rangle(t, r)$ converged very well for $k = 72$. $\langle\beta_1\rangle(t, r)$ converges later with the number of samples taken into account, since distributions are computed from fewer persistent homology classes with corresponding properties. Yet, additional samples do not alter the overall shape of $\langle\beta_1\rangle(t, r)$ anymore, solely reducing occurring statistical fluctuations.

As observed in Sec. 5.1, the average maximum death radius, $\langle r_{d,1,\max}\rangle(t)$, is a quantity that is very sensitive to particular geometric arrangements of points in analyzed point clouds. Resembling this effect, in Fig. 18 we display $\langle r_{d,1,\max}\rangle(t)$ for different $n$ and $\bar{\nu}$. Clearly, occurring oscillations drastically reduce with $k$ increasing. Up to a few outliners, regions of approximate power-law behavior converged properly for $k = 72$ as studied in the main text.

To sum up, different persistent homology observables converge differently fast with the number of classical-statistical samples, $k$, taken into account in averaging. Corresponding differences among their convergence behavior can be easily understood geometrically.

# G   Numerical protocol to extract persistent homology scaling exponents

Key to the analysis of results in our nonrelativistic Bose gas testbed in Sec. 3.4 is the extraction of persistent homology scaling exponents from approximately self-similar birth and death radii distributions. This appendix serves as a description of the applied protocol to accomplish this task.

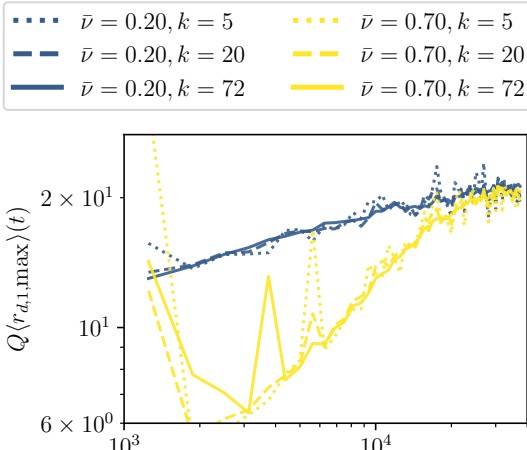

Figure 18: The average maximum death radius of 1-dimensional persistent homology classes varying with time, displayed for $\bar{\nu}$-values and numbers of samples to average, $k$, as indicated.

We first define rescaled variants of the birth and death radii distributions,

$$\langle \mathcal{B}_\ell \rangle^{\text{resc}}(t, r_b) = (t/t')^{\eta_2 - \eta_1'} \langle \mathcal{B}_\ell \rangle(t, (t/t')^{-\eta_1} r_b), \tag{G.1a}$$

$$\langle \mathcal{D}_\ell \rangle^{\text{resc}}(t, r_d) = (t/t')^{\eta_2 - \eta_1} \langle \mathcal{D}_\ell \rangle(t, (t/t')^{-\eta_1'} r_d). \tag{G.1b}$$

Distributions at later times are compared with those at the reference time $t'$, chosen to be the time at which the self-similar evolution sets in. However, we could equally well have chosen any other reference time within the self-similar scaling regime. Denote by $t_k > t'$, $k = 1, \ldots, N_{\text{com}}$, all corresponding comparison times. If birth and death radii distributions were evolving perfectly self-similar following Eqs. (19a) and (19b), we would find

$$\Delta \langle \mathcal{B}_\ell \rangle(t, r_b) = \langle \mathcal{B}_\ell \rangle^{\text{resc}}(t, r_b) - \langle \mathcal{B}_\ell \rangle(t', r_b) = 0, \tag{G.2a}$$

$$\Delta \langle \mathcal{D}_\ell \rangle(t, r_d) = \langle \mathcal{D}_\ell \rangle^{\text{resc}}(t, r_d) - \langle \mathcal{D}_\ell \rangle(t', r_d) = 0. \tag{G.2b}$$

Numerically, even for the correct triple of exponents $(\eta_1, \eta_1', \eta_2)$ this is only approximately true due to statistical uncertainties as well as systematic errors entering since systems typically only enter the vicinity of a nonthermal fixed point. We optimize scaling exponents by means of minimizing occurring deviations, quantified by

$$\chi^2(\eta_1, \eta_1', \eta_2) = \chi_b^2(\eta_1, \eta_1', \eta_2) + \chi_d^2(\eta_1, \eta_1', \eta_2), \tag{G.3a}$$

$$\chi_b^2(\eta_1, \eta_1', \eta_2) = \frac{1}{N_{\text{com}}} \sum_{k=1}^{N_{\text{com}}} \frac{\int_{r_{\min}}^{r_{\max}} dr_b \, \Delta \langle \mathcal{B}_\ell \rangle(t_k, r_b)^2}{\int_{r_{\min}}^{r_{\max}} dr_b \, \langle \mathcal{B}_\ell \rangle(t', r_b)^2}, \tag{G.3b}$$

$$\chi_d^2(\eta_1, \eta_1', \eta_2) = \frac{1}{N_{\text{com}}} \sum_{k=1}^{N_{\text{com}}} \frac{\int_{r_{\min}}^{r_{\max}} dr_d \, \Delta \langle \mathcal{D}_\ell \rangle(t_k, r_d)^2}{\int_{r_{\min}}^{r_{\max}} dr_d \, \langle \mathcal{D}_\ell \rangle(t', r_d)^2}. \tag{G.3c}$$

Lower and upper limits of integration in the appearing expressions are set to $Qr_{\min} = 1.5$ and $Qr_{\max} = 25.0$ for all $\bar{\nu} \leq 0.7$ and $Qr_{\min} = 1.0$ and $Qr_{\max} = 10.0$ for $\bar{\nu} = 0.8$. A priori, the given expressions for $\chi_{b/d}^2(\eta_1, \eta_1', \eta_2)$, are equally sensitive to the behavior at all scales of radii, increasing the weight of data points whose deviations are large. Linear interpolations

are employed to obtain birth and death radii distributions at rescaled birth and death radii, respectively.

Minimizing deviations as measured by $\chi^2(\eta_1, \eta_1', \eta_2)$, the optimal triple $(\tilde{\eta}_1, \tilde{\eta}_1', \tilde{\eta}_2)$ is obtained. Analogously to Refs. [32, 63], a likelihood function is defined as

$$W(\eta_1, \eta_1', \eta_2) = \frac{1}{\mathcal{N}} \exp\left( -\frac{\chi^2(\eta_1, \eta_1', \eta_2)}{2\chi^2(\tilde{\eta}_1, \tilde{\eta}_1', \tilde{\eta}_2)} \right), \tag{G.4}$$

$\mathcal{N}$ being a normalization constant such that

$$\int d\eta_1 \, d\eta_1' \, d\eta_2 \, W(\eta_1, \eta_1', \eta_2) = 1. \tag{G.5}$$

Marginal likelihood functions are obtained upon integrating over two of the exponents, for instance,

$$W(\eta_1) = \int d\eta_1' \, d\eta_2 \, W(\eta_1, \eta_1', \eta_2). \tag{G.6}$$

We fit marginal likelihood functions with Gaussian distributions to estimate corresponding standard deviations, $\sigma_{\eta_1}, \sigma_{\eta_1'}$ and $\sigma_{\eta_2}$, the means still being given by $\tilde{\eta}_1, \tilde{\eta}_1'$ and $\tilde{\eta}_2$.

To derive time-dependent persistent homology scaling exponents, we apply the described fitting procedure with a fixed reference time $Qt'$ for $N_{\text{com}} = 3$ times, simultaneously: $Qt_{\text{min}}$ as indicated in the main text as well as $Qt_{\text{min}} + 625$ and $Qt_{\text{min}} + 1250$. Repeating this procedure for different $Qt_{\text{min}}$, we obtain time-dependent scaling exponents.

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
