# Peer review of "Finding self-similar behavior in quantum many-body dynamics via persistent homology"

_SciPost Physics, doi:SciPost Phys. 11, 060 (2021)_

## Round 1 · Referee Report · Anonymous (Referee 1) · 2021-2-10

Strengths

1) visionary 2) cross-disciplinary 3) bold

Weaknesses

1) hard to read 2) full of jargon 3) weak interpretation

Report

By reviewing the manuscript by Spitz et al. I developed a number of contrasting feelings about their work.

Their starting motivation seem to extend topological data methods to the analysis of quantum experiments. This is certainly original and they end up delivering a manuscript which is very much trans-disciplinary and certainly highly original.

The first issue is that such originality is perhaps too far reaching. By admission of the authors, their explanation on the continuous scaling is highly conjectural (mid of page 14) and motivation for future studies or even their original intention is hard to find (again, the authors are honest about it — see final paragraphs of the conclusions).

Combined with a strongly jargon language and some cryptic passages (see below), all this risks to prevent the interested or curious reader from benefiting of their results and potentially expanding such research line.
So I believe it should be in the own interest of the authors to significantly improve the manuscript.

I now report a number of instances mostly taken from the first ten pages, which carry on to the entire article:

— the intro is too technical; I understand they have a summary of homologies later on, but it is quite discouraging (the reader is bombarded with technical words from algebraic topology and would tend to skip the first two pages)

— sec. 2.1: what is the classical-statistical approximation? (I am aware they reference) why Q is used several lines before properly being introduced? the factor ’50’ in eq (1) appears weird — why not using a tunable parameter? this section is overall fine but this additional imprecisions increase the sense of discouragement in keep going with the manuscript

— hard/cryptic paragraphs persist even in section 2.2 which should help the reader: for instance at page 6 the paragraph that starts with ‘The construction of persistent homology groups etc etc’

— these birth and death radii should be connected with something physical, otherwise the entire original motivation is lost

— I am not sure how much the reader can follow on non-thermal fixed points and turbulence if he/she is disconnected from previous results (I am aware the authors work on this since several years)

It really seems sometimes that the paper has not been proof-read before submission. The five examples above could serve as a guideline to massively improve the presentation.

Two concerns about the main result:

— it is really destablising to talk about scaling exponents that depend on non-universal parameters like the filtration and Qt_min — can the authors mitigate this in the revised version of the manuscript?

— is there anything special about the model they consider? does this scaling of the homologies depend only on dimensionality, symmetries, and other few relevant quantities as in conventional scaling theory in stat mech?
Conjectural thoughts would already help to put all this in proper context

In conclusion: the work might definitely help the community to take a new angle in solving the many-body problem with science data/advanced math, however, at the moment the presentation is borderline with being not comprehsible and it would be a pity (in primary instance, for the authors themselves).

Requested changes

1) review the presentation following the criteria in the report 2) comment on universality of the results

---

## Round 1 · Referee Report · Anonymous (Referee 2) · 2021-4-26

Strengths

1) Nice introduction to topological data analysis techniques

2) Clear discussion of the performed numerical simulations

3) Potential importance for future studies of experimental data in ultracold atoms systems with topological data analysis

Weaknesses

1) Not enough discussed how the method can be used to treat general, strongly interacting QFTs

2) Not enough motivated the claim that "a refined classification of nonequilibrium universal phenomena" is (or can be) obtained.

Report

The paper introduces topological data analysis techniques, introducing the so-called "persistent homology observables". After such an introduction, the authors considered data from a classical-statistical simulation of a 2D Bose gas far from equilibrium. Birth and death radii distributions are introduced and numerical results discussed.

The idea of using topological data analysis is certainly interesting, and motivates me to give an overall positive evaluation. Also appreciated by me has been the clarity with which the numerical simulations are presented and discussed.

The point which at variance leaves me skeptical is the particular system studied. To reach and motivate the ambitious goal the authors set, I feel they should/could have discussed (before their example or even instead of) a simpler, and in my opinion, clearer example: a system slightly perturbed from the critical point, and studying the critical dynamics with their methods. I would have expected to see this in a mean-field model, to see what their machinery would have produced. In that case the authors could have used a description based on Gross-Pitaevskii or time-dependent Ginzburg-Landau, and compared with known results, including how the obtained scaling exponents (of birth and death radii distributions) would have been related to the known dynamical critical exponent z, at least for some of the Hohenberg-Halperin classes. That could have be applied to a Bose-Hubbard model near the superfluid-Mott transition, or even to a Bose-Einstein condensate (without optical lattice) near the critical point. How/what the "scaling exponents" introduced in the paper would have been? It is possible that I may have missed literature doing topological data analysis on these simplified models, but in case such results already exist (or maybe the authors have already worked out) I would have appreciated very much an introductive discussion on this point.

The authors instead decided to go for the 2D Bose gas, and there one expects strong deviations from mean-field. It is not even clear to me whether with their approach they can reproduce the equation of state [Phys. Rev. Lett. 87, 270402 (2001)] and the highly non-perturbative parameter in it. In the regime in which they put themselves, the classical-statistical approximation, expectation values of quantum observables are computed as ensemble-averages of classical field configurations: so, one may raise the question of how the approach would work when classical-statistical approximation fails. And, similarly, what should be the procedure to compare with data extracted from the experiment when classical-statistical approximation fails? One answer to this objection could be that one should start form the simplest case before, but then we are back to my previous question on why simpler setups (such a weakly interacting Bose gas near its critical temperature) have been not discussed before. From my point of view even the 1D Bose gas would have better, but this is - I admit - certainly matter of taste.

Finally, I am bit surprised by the emphasis the authors give on the "scaling exponents". Indeed, analyzing Monte Carlo snapshots of Ising models (or percolation) one could define critical exponents from peculiar non-trivial subset one can construct from them, such as the Fortuin-Kasteleyn representation. Also there one can define appropriate critical exponents, the real question being how they are related to the other critical exponents.

In conclusion, I liked the idea of the paper. I feel that a more physical introduction to the problem, and in particular a discussion on mean-field-like examples [or near T_BEC, where experimental data could be worked out] would have been important and would have considerably improved both the clarity of the paper and the significance of the presented method.

Requested changes

In the report I mentioned several points which in my opinion could be discussed better. I do understand that too much material cannot be added (or changed) from the present version, but my suggestion is to consider the previous comments and accordingly improve the discussion, possibly including of what the topological data analysis machinery would produce near a critical point (at least in mean-field). Even if this discussion would be very simple with respect to the rich phenomelogy of the 2D Bose gas, I think it would help the clarity of the paper.

---

## Round 2 · Referee Report · Anonymous (Referee 1) · 2021-6-22

Report

The authors have taken into account large part of the comments in the previous round of Referral and they have done an effort in improving the quality of their presentation, which was one of the main critical aspects. I am glad to accept their work for publication in SciPost.

---

## Round 2 · Referee Report · Anonymous (Referee 2) · 2021-8-26

Report

The authors took care of many of the comments and points raised by the referees. I have to say that, despite this effort, I continue to think that the method could have better explained in relation to critical phenomena (and/or to systems "simpler" than the ones studied). Put in another way, I feel that many other interesting things may follow from the method presented in this paper. However, the method presented is interesting and the clarity considerably improved, so I am favour of the publication of the paper.

---

## Round 2 · Author Response

Detailed replies to comments made by the first referee

1) “By admission of the authors, their explanation on the continuous scaling is highly conjectural (mid of page 14) and motivation for future studies or even their original intention is hard to find (again, the authors are honest about it — see final paragraphs of the conclusions).”
a) It is known that nonthermal fixed points and the related scaling behavior can encompass very different theories. For instance, the scaling properties in the infrared momentum regime of relativistic and nonrelativistic scalar field theories agree [Phys. Rev. D 92 (2015)]. These apparent similarities need to be tested against refined classification schemes. As mentioned in the introduction of our manuscript, it is a key motivation for our work to search for suitable quantities which both capture relevant properties of nonthermal fixed points and provide novel degrees of freedom to study them. Particularly, in light of the found scaling exponent spectrum, our results constitute a potentially enriching step in that new direction, the filtration function providing a new degree of freedom in order to discriminate between field-theoretic information under study.

2) “Combined with a strongly jargon language and some cryptic passages (see below), all this risks to prevent the interested or curious reader from benefiting of their results and potentially expanding such research line.”
a) In the revised manuscript we included an updated introduction which contains less mathematical terms and should thus be better accessible to a readership educated in physics. We hope that the included updates of the section which introduces persistent homology to the reader can further support the readability.

3) “[T]he intro is too technical; I understand they have a summary of homologies later on, but it is quite discouraging (the reader is bombarded with technical words from algebraic topology and would tend to skip the first two pages)”
a) We updated the introduction, accordingly, making it more accessible to non-mathematicians and only including mathematical jargon wherever crucial.

4) “[S]ec. 2.1: what is the classical-statistical approximation? (I am aware they reference) why Q is used several lines before properly being introduced? the factor ’50’ in eq (1) appears weird — why not using a tunable parameter? this section is overall fine but this additional imprecisions increase the sense of discouragement in keep going with the manuscript”
a) In the revised version, the classical-statistical approximation is in brevity described in the section on self-similarity in occupation numbers. Valid equivalently, in the revised manuscript we change $Qt=0$ to $t=0$, not employing the scale $Q$ before introducing it later. Furthermore, in the new manuscript version we introduced the chosen initial conditions more carefully, mentioning the chosen initial momentum space box in occupation numbers, keeping the amplitude $f_0$ more general and subsequently only restricting to a value of $f_0=50/(2mGQ)$.

5) “[H]ard/cryptic paragraphs persist even in section 2.2 which should help the reader: for instance at page 6 the paragraph that starts with ‘The construction of persistent homology groups etc etc’”
a) We updated the paragraph on the robustness of the employed computational approach to include less mathematical terms which have not been introduced properly before (revised manuscript, Sec. 3.1). Admittedly, the remaining text passages of this section should already be amenable to a broad readership with physics background, given that we introduce the notion of filtration functions with care and discuss in detail the physical degrees of freedom included in point clouds of different filtration parameters.

6) “[T]hese birth and death radii should be connected with something physical, otherwise the entire original motivation is lost”
a) Persistent homology provides a means to study connectivity structures present in point clouds in a robust fashion which is insensitive to noise in the data due to the topological constructions undertaken. It is a crucial feature of persistent homology to be able to discriminate between topological structures of different physical length scales involved. The construction of point clouds as sublevel sets of field amplitudes allows for a discrimination between vortices and bulk excitations – a distinction known to discriminate between crucial dynamical degrees of freedom in the two-dimensional Bose gas [New J. Phys. 19, 093014 (2017)]. Our topological data analysis pipeline with distributions of birth and death radii as the central objects under investigation features well-interpretable geometric objects – loops of different dimensions, shapes, and sizes – contained in easily interpretable data which is even experimentally accessible as we elaborate on in a paragraph on the applicability of our methods to optical density images.

7) “I am not sure how much the reader can follow on non-thermal fixed points and turbulence if he/she is disconnected from previous results (I am aware the authors work on this since several years)”
a) In the revised manuscript we included a section which discusses nonthermal fixed points and the scaling of two-point functions in the two-dimensional Bose gas, moving the content of Appendix F to the main text. This should help the reader not familiar with nonthermal fixed points understand the core messages of our work.

8) “[I]t is really destab[i]lising to talk about scaling exponents that depend on non-universal parameters like the filtration and Qt_min — can the authors mitigate this in the revised version of the manuscript?”
a) The time-dependent scaling exponent analysis carried out to obtain Fig. 7 in our manuscript reveals two key findings: A plateau for $\bar{\nu} < 0.5$ which is constant in time and the presence of a peak which comparably slowly shifts gradually towards higher filtration parameters $\bar{\nu}$. Both these findings persist in time and as such are to be regarded independent of the chosen value of $Qt_\min$. Solely the precise position of the peak changes in the course of time and is to be studied via time-dependent scaling exponents. The entire phenomenological discussion of the scaling exponent spectrum in our work is based on these two findings and not on the precise peak position. As such, we regard the time-dependency of the scaling exponents as a mere technicality.
b) The dependence on the filtration function is from our point of view not a mere technicality but instead a feature of our analysis. Persistent homology is robust in the sense that slightly varying the filtration function, the obtained results are nearly unaltered. However, one may choose completely different filtration functions such as phases computed from the complex-valued field configurations instead of amplitudes. We cannot expect these very different filtration functions to give rise to the same self-similar behavior of persistent homology observables. Crucially, via the choice of a filtration function one can study length-scale resolved connectivity structures from selected field-theoretic information.
c) Given that our data is still well described by a self-similar scaling ansatz, we carry on regarding the corresponding exponents as scaling exponents. Nonetheless, in the revised manuscript at multiple positions in the text and in the title, we have been more careful about connecting the found scaling exponents with universality far from equilibrium. The latter certainly demands more attention beyond the scope of the present manuscript, such as investigating the dependence on initial conditions.

9) “[I]s there anything special about the model they consider? does this scaling of the homologies depend only on dimensionality, symmetries, and other few relevant quantities as in conventional scaling theory in stat mech?
Conjectural thoughts would already help to put all this in proper context.”
a) Field-theoretically, the single-component nonrelativistic Bose gas in two spatial dimensions is the simplest system which can be both analysed reasonably by persistent homology techniques and shows nonthermal fixed points [New J. Phys. 19, 093014 (2017)]. For instance, to find nonequilibrium self-similar dynamics in a single spatial dimension, internal degrees of freedom are necessary [Phys. Rev. A 99, 033611 (2019)].
b) The plateau in observed scaling exponents at low filtration parameters with exponent values of approximately 0.2 can be attributed to anomalous vortex dynamics [New J. Phys. 19, 093014 (2017), SciPost Phys. 8, 039 (2020)]. Being topological defects of field configurations in position space, vortices, and their dynamics in general strongly depend on dimensionality. Defect-dominated nonthermal fixed points are thus expected to depend on dimensionality, symmetries, and the field-content under study. On the other hand, nonthermal fixed points can show similar scaling behavior across dimensions and symmetry groups due to waves interacting nonlinearly [Phys. Rev. D 92, 025041 (2015), Phys. Rev. D 101, 091902 (2020)]. The concrete role of topological defect and nonlinear wave dynamics in nonthermal fixed point behavior is still to be untangled, conclusively. Our work shows that persistent homology observables can provide novel degrees of freedom which can be of use here.

Detailed replies to comments made by the second referee

Comments on the denoted weaknesses:
1) “Not enough discussed how the method can be used to treat general, strongly interacting QFTs”
a) See the answer to point 4 below.

2) “Not enough motivated the claim that "a refined classification of nonequilibrium universal phenomena" is (or can be) obtained.”
a) With the novel Sec. 2 of the revised manuscript we gladly incorporate a more careful description of the self-similar scaling via the well-established occupation number spectrum, in particular showing that a single pair $(\alpha,\beta)$ of scaling exponents allows for a successful rescaling of the latter. Clearly, the obtained persistent homology scaling exponent spectrum contains more structure, as such providing from our point of view a sound basis for the above claim. In particular, given the freedom to choose a filtration function, the computational pipeline provides further novel quantities to study self-similar phenomena far from equilibrium, possibly allowing for untangling relevant degrees of freedom. We interpret the presence of a peak in the deduced scaling exponent spectrum as a signature in favor of the suitability of persistent homology observables for that purpose.

Detailed answers to points included in the report:

1) “The point which at variance leaves me sceptical is the particular system studied. To reach and motivate the ambitious goal the authors set, I feel they should/could have discussed (before their example or even instead of) a simpler, and in my opinion, clearer example: a system slightly perturbed from the critical point, and studying the critical dynamics with their methods. I would have expected to see this in a mean-field model, to see what their machinery would have produced. In that case the authors could have used a description based on Gross-Pitaevskii or time-dependent Ginzburg-Landau, and compared with known results, including how the obtained scaling exponents (of birth and death radii distributions) would have been related to the known dynamical critical exponent z, at least for some of the Hohenberg-Halperin classes. That could have be applied to a Bose-Hubbard model near the superfluid-Mott transition, or even to a Bose-Einstein condensate (without optical lattice) near the critical point.”
a) The critical behavior under investigation is to be distinguished from critical behavior in equilibrium. In our work the self-similar scaling of relevant observables unfolds in the course of time and not in the immediate vicinity of a critical temperature. In addition, to reach the vicinity of nonthermal fixed points no fine-tuning of (order) parameters is necessary. Nonthermal fixed points even form nonequilibrium attractors of time-evolutions of quantum many-body systems encompassing different types of initial conditions [Phys. Rev. D 89, 114007 (2014), Nature 563, 217-220 (2018)]. Nonthermal fixed points being comparably well studied with clear phenomenological signatures (self-similar scaling), they provide an ideal testing ground for persistent homology observables in dynamical quantum many-body systems.
b) Field-theoretically, the single-component nonrelativistic Bose gas in two spatial dimensions is the simplest system which can be both analysed reasonably by persistent homology techniques and shows nonthermal fixed points [New J. Phys. 19, 093014 (2017)]. For instance, to find nonequilibrium self-similar dynamics in a single spatial dimension, internal degrees of freedom are necessary [Phys. Rev. A 99, 033611 (2019)].
c) Taking different approaches, in recent years persistent homology has also been applied to critical behavior in equilibrium [Phys. Rev. E 93, 052138 (2016), Phys. Rev. Research 2, 043308 (2020)], with the main goal of topologically identifying phase transitions and providing novel degrees of freedom to capture order in physical systems with more complicated degrees of freedom. We included a description of the main available papers describing equilibrium models in the introduction. Our approach should be easily adaptable to the thermal equilibrium case, which, however, lays beyond the scope of the present paper.

2) “How/what the "scaling exponents" introduced in the paper would have been? It is possible that I may have missed literature doing topological data analysis on these simplified models, but in case such results already exist (or maybe the authors have already worked out) I would have appreciated very much an introductive discussion on this point.”
a) Far from equilibrium our study is the first one investigating scaling behavior of persistent homology observables in quantum many-body systems. Even in equilibrium there has been to the best of our knowledge no study which analysed persistent homology observables with regard to scaling phenomena. As such, we expect no such discussion to be present in the literature already. Solely, we can again refer to studies in equilibrium which qualitatively investigated phase transitions [Phys. Rev. E 93, 052138 (2016), Phys. Rev. Research 2, 043308 (2020)].

3) “The authors instead decided to go for the 2D Bose gas, and there one expects strong deviations from mean-field. It is not even clear to me whether with their approach they can reproduce the equation of state [Phys. Rev. Lett. 87, 270402 (2001)] and the highly non-perturbative parameter in it.”
a) As mentioned already, the two-dimensional single-component Bose gas is among the simplest models to both reveal self-similar dynamics far from equilibrium and give rise to at least one-dimensional persistent homology classes with non-zero birth radii. The persistence diagram of zero-dimensional homology classes would just amount to a descriptor of inter-point distances in point clouds. One- and higher-dimensional persistent homology classes instead contain topological information on loops and voids, etc.
b) The classical-statistical approximation is valid for large occupation numbers and small couplings [Phys. Rev. Lett. 88, 041603 (2002), Phys. Rev. A 76, 033604 (2007)]. Quantum thermal equilibrium is not amenable to a classical-statistical description. It is due to this fact that our approach will not be able to reproduce the equation of state referenced by the referee. Still, via the classical-statistical description with the Gross-Pitaevskii equation at its heart the dynamical evolution away from and towards the BKT phase transition can be studied [Phys. Rev. A 86, 013624 (2012)].

4) “In the regime in which they put themselves, the classical-statistical approximation, expectation values of quantum observables are computed as ensemble-averages of classical field configurations: so, one may raise the question of how the approach would work when classical-statistical approximation fails. And, similarly, what should be the procedure to compare with data extracted from the experiment when classical-statistical approximation fails?”
a) The classical-statistical approximation covers a wide variety of physically interesting scenarios and phenomena [Phys. Rev. Lett. 88, 041603 (2002), Phys. Rev. A 76, 033604 (2007)]. As such, our topological data analysis pipeline is already quite generally applicable to classical systems, as well as quantum systems in the classical-statistical regime. Furthermore, in the section on point cloud phenomenology we speculate about possible applications to experimental data. Optical density images providing absolute squares of nonrelativistic field values, our approach can be employed directly as it is. In principle, our approach can be applied as well to field configurations obtained from importance-based sampling techniques such as Monte Carlo simulations and as such extends widely beyond the classical-statistical regime.

5) “From my point of view even the 1D Bose gas would have better, but this is - I admit - certainly matter of taste.”
a) Nonequilibrium self-similar dynamics in a single spatial dimension requires internal degrees of freedom, else, kinetic arguments prohibit necessary scattering processes for the redistribution of particles from taking place. This would complicate the entire topological analysis.

6) “Finally, I am bit surprised by the emphasis the authors give on the "scaling exponents". Indeed, analyzing Monte Carlo snapshots of Ising models (or percolation) one could define critical exponents from peculiar non-trivial subset one can construct from them, such as the Fortuin-Kasteleyn representation. Also there one can define appropriate critical exponents, the real question being how they are related to the other critical exponents.”
a) Given the robust observation of self-similar scaling in persistent homology observables, it is natural to talk about scaling exponents as the exponents appearing in the self-similar scaling ansatz to the asymptotic persistence pair distribution. Discussing scaling exponents, we a priori do not assume them to be universal, i.e., including different physical systems, dimensions, quantities and initial conditions. In the revised manuscript we have been more careful in distinguishing the mere observation of scaling with the involved scaling exponents from possible universal aspects of it.

7) “[M]y suggestion is to consider the previous comments and accordingly improve the discussion, possibly including of what the topological data analysis machinery would produce near a critical point (at least in mean-field). Even if this discussion would be very simple with respect to the rich phenomenology of the 2D Bose gas, I think it would help the clarity of the paper.”
a) We gladly included in the introduction a brief discussion of equilibrium works present in the literature and moved the discussion of scaling of two-point functions in the system from the Appendix to a new section directly after the introduction. There we also clarify nonthermal fixed point characteristics, providing a more pedagogical introduction to self-similar scaling behavior in the vicinity of nonthermal fixed points and strengthening the motivation for persistent homology observables.

---

## Round 2 · List of Changes

• Updated title, focussing more on self-similarity instead of universality
  • Updated abstract slightly
  • Changed beginning and later paragraphs of the introduction, such that it becomes more accessible to non-mathematicians, in addition focussing as well more on self-similarity instead of universality. Information on available papers studying equilibrium phases with persistent homology methods has been added.
  • Introduced a novel Sec. 2, which discusses nonthermal fixed points in the well-established occupation number spectrum. Therein, the chosen initial conditions are introduced more carefully than before.
  • Revised manuscript, Sec. 3.1, includes less mathematical terms not introduced properly before.
  • Tiny changes throughout the work in order to discriminate more clearly between the observation of self-similarity and possible universal aspects of it.

---

## Editorial Decision

published